# DecompRL: Solving More Problems with Less Tokens

## Abstract

While repeated sampling from Large Language Models (LLMs) is a robust baseline for competitive programming and other automatically verifiable problems, it comes at a steep GPU cost. Reinforcement learning (RL)-based post-training can reduce the necessary sample size, but often worsens generations diversity, which limits performance in the large-scale sampling regime. Online RL is itself bottlenecked by the performance of the starting policy and the heavy compute required for inference. We introduce DecompRL, an algorithm inspired by modular inference that trains policies to decompose complex problems into separate, parallelizable functions. By recombining these modules into polynomially many solutions, DecompRL shifts the RL bottleneck from GPU-based inference to CPU-based evaluation. This enables massive scaling at a fraction of the cost, improving sparse reward discovery and solving complex problems that remain out of reach for standard RL.

## 1 Introduction

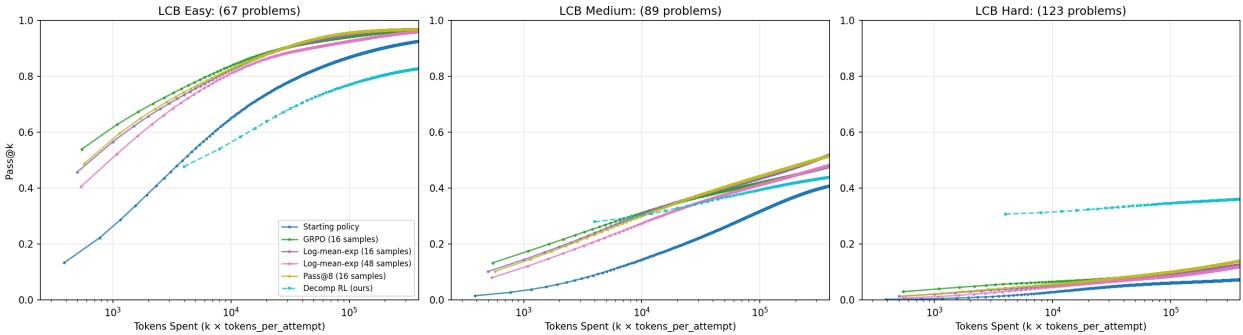

Figure 1: **Decomp RL solves harder problems than existing RL methods.** Our hierarchical training learns 1) a decomposition policy that breaks down problems into simpler functions and 2) an implementation policy that writes code for each function in parallel. Given a high token budget, Decomp RL solves up to 35% problems of the LiveCodeBench hard subset (2024/08/01 to 2025/02/01) Jain et al. (2024a).

Large language models (LLMs) have demonstrated remarkable prowess in automatically verifiable domains like competitive programming. Currently, the dominant strategy for state-of-the-art performance relies on repeated sampling—generating numerous potential solutions to pass a verifier: the pass@k regime (Chen et al., 2021). However, the major limitation of this approach lies in its significant GPU computational costs for scaling inference. Attempts to mitigate this through reinforcement learning post-training have primarily focused on improving single-sample accuracy (pass@1) at the cost of pass@k in high sampling regimes for $k >> 100$. This reduction in diversity undermines its effectiveness when scaling inference at test time, because performance gains grow logarithmically for linear computational costs (Appendix 10).

We propose DecompRL: a reinforcement learning framework that fundamentally shifts the scaling bottleneck from expensive GPU inference to cheap CPU evaluation. Inspired by modular inference strategies (Zelikman et al., 2023), DecompRL trains models to decompose complex challenges into a hierarchy of independently

solvable sub-functions. For example, for a sorting task instead of generating a single monolithic solution, the model identifies $N$ sub-problems—such as partitioning, merging, and base-case sorting. By generating $K$ implementations for each of the $N$ modules in parallel, we can recombine them into up to $K^N$ unique candidate solutions. This approach creates a combinatorial explosion of potential programs for a fixed inference budget of only $K \times N$ tokens, allowing us to scale search on cheap CPU clusters without increasing the GPU footprint.

By framing hierarchical modular inference as an RL objective, DecompRL prioritizes exploration (Section 4.1), maximizes the utility of recombinations (Section 4.2) and successfully discovers sparse rewards (Section 4.3). DecompRL demonstrates that solving harder problems does not strictly require more GPU tokens—it requires smarter, modular structures.

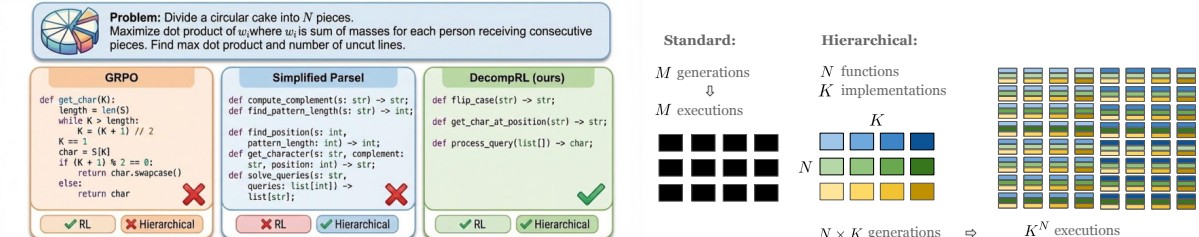

Figure 2: **Learning to scale CPU budget. (left)** Example problem from LiveCodeBench and the generated solutions with regular and hierarchical inference. **(right)** Solving a problem directly in $M$ tries leads to $M$ complete code solutions to evaluate. Hierarchical inference instead generates $N$ functions each implemented $K$ times independently. Recombining these partial code blocks gives up to $K^N$ full code solutions to evaluate.

## 2 Method

We introduce DecompRL: a reinforcement learning recipe to learn hierarchical generation (Section 2.1) as described in Parsel (Zelikman et al., 2023) and scale inference-time compute (Section 2.2).

### 2.1 Hierarchical inference

We consider the task of generating code in a modular fashion, where model users have tighter control over the generation process than in the standard setup of whole-code generation from an autoregressive language model.

Standard whole-code generation models the joint distribution of a piece of code of multiple parts $I_1, \ldots, I_n$ by means of its autoregressive decomposition and can optionally include a plan $D$ sampled before $I_1$. Surpressing conditionning on the problem $x$, we have:

$$\pi(I_1, \ldots, I_n) = \pi(I_1|D) \prod_{i=1}^{n} \pi(I_i \mid I_{i-1}, \ldots, I_1, D).$$

In this work, we instead consider a hierarchical decomposition:

$$\pi(D, I_1, \ldots, I_n) = \pi(D) \prod_{i=1}^{n} \pi(I_i \mid D)$$

which models the implementations of the pieces $I_i$ as conditionally independent of each other given $D$, and captures all interaction in the *decomposition plan $D$*.

In practice where we consider code generation tasks in Python, we represent decompositions $D$ by a set of functions specified by their signatures and natural language specifications in the form of docstrings. The parts $I_i$ are then *implementations* of those functions. We give an example of a decomposition in Figure 2 and of a full hierarchical trajectory $(D, I_1, I_2)$ in the Appendix 19. We use prompted and trained language models for the decomposition and implementation policies $\pi(D)$ and $\pi(I_i \mid D)$ (see Appendix 16 for full prompts).

## 2.2 Recombination

Hierarchical inference comes with an important benefit: while the conditional independence assumption makes individual samples less precise, it can be exploited to generate samples cheaply by recombination. Here and in the following, let $D$ be a decomposition of size $n$ and $I_i^j, 1 \leq i \leq n, 1 \leq j \leq k$, be $k$ implementations per function $I_i$ described by $D$. From these $nk$ implementation samples, we can create $k^n$ complete trajectories by considering all combinations $(D, I_1^{j_1}, \ldots, I_n^{j_n})$ for $1 \leq j_1, \ldots, j_n \leq k$. In other words, the number of complete trajectories scales *polynomially* in $k$ while the inference cost scales linearly.

This is particularly useful in domains where rewards are only observed for complete trajectories, which is the case for *verifiable reasoning domains* like mathematics and code. In such setups, verification is a cheap CPU operation while generation is an expensive GPU operation involving large language models. In this work, we consider the task of competitive programming where the model must produce a code solution to a problem in natural language. The reward is 1 if the code passes all private tests (unknown to the model at inference time) and 0 otherwise.

## 2.3 Reinforcement learning: policy gradient estimators

As customary with large language models, we consider policy gradient reinforcement learning algorithms (Williams, 1992). We write $\tau = (D, I_1, \ldots, I_n)$ for a trajectory, $r(\tau)$ for the reward it receives and $\pi_\theta$ as the policy given by a large language model. According to the policy gradient theorem (Sutton & Barto, 1998), an unbiased estimate of the gradient $g$ of the expected reward is given by

$$g = \nabla_\theta \mathbb{E}_{\tau \sim \pi_\theta}[r(\tau)] = \mathbb{E}_{\tau \sim \pi_\theta}\left[\sum_{a \in \tau}(r(\tau) - b)\nabla_\theta \log \pi_\theta(a)\right],$$

for any baseline $b$ which does not depend on the action $a \in \{D, I_1, \ldots I_n\}$.

We can now view *recombination* as a way to produce Monte-Carlo estimates of the policy gradient for hierarchical inference more efficiently. Given $k$ implementations $I_i^j$ of the functions $I_i$ in the decomposition $D$, the standard estimator for $g$ used in sequential reinforcement learning is

$$\hat{g}_{\text{standard}} = \frac{1}{k}\sum_{j \leq h}\sum_{a \in \{D, I_1^j, \ldots, I_n^j\}}(r(D, I_1^j, \ldots, I_n^j) - b)\nabla_\theta \log \pi_\theta(a),$$

which is optimal in the case of linear (autoregressive) rollouts where $I_k$ depends on $I_j$ for $k > j$.

For hierarchical inference, however, we made the *modeling assumption* that implementations are conditionally independent and all interactions are captured by their dependence on the decomposition $D$. In this case, we can evaluate all combinations of implementations generating a matrix $(I_i^j)_{j,i} \, \forall i, j$ for al functions. We can obtain the improved estimator:

$$\hat{g}_{\text{hierarchical}} = \frac{1}{k^n}\sum_{j_1, \ldots, j_n \leq k}\sum_{a \in \{D, I_1^{j_1}, \ldots, I_n^{j_n}\}}(r(D, I_1^{j_1}, \ldots, I_n^{j_n}) - b)\nabla_\theta \log \pi_\theta(a),$$

of which $\hat{g}_{\text{standard}}$ represents the diagonal terms $(I_i^j)_{i,i} \, \forall i$. Clearly, both estimators are unbiased, but we have $\text{Var}(\hat{g}_{\text{hierarchical}}) \leq \text{Var}(\hat{g}_{\text{standard}})$, as we show in Theorem 9.2.

## 2.4 Reinforcement learning objectives

We do not expect hierarchical inference to improve upon whole code generation when compared on a single-attempt (pass@1) basis. This is because hierarchical inference comes with a loss in precision (conditionally independence instead of joint modeling of implementations) and with only one sample we do not gain anything from the recombination. Instead, we propose to focus on the task of efficient inference scaling, i.e, comparing pass@k (Chen et al., 2021) metrics for large values of $k >> 100$.

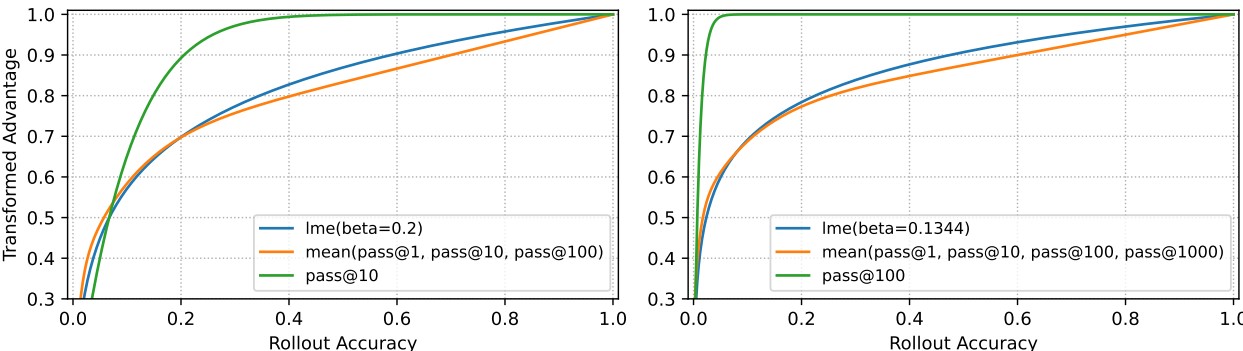

Figure 3: Nonlinear reward transformations with utility functions. Reward transformation induced by $\text{logmeanexp}_\beta$ for different values of $\beta$, alongside the ones of pass@k: $u(p) = 1 - (1 - p)^k$, and the mean of pass@k values. See Appendix 11 for details.

Given $n$ rollouts $\tau_j$ with rewards $r = (r(\tau_1), \ldots, r(\tau_n))$ and a *multi-sample objective function* $f(r)$, Tang et al. (2025) optimize $E_{\tau_i \sim \pi_\theta}[f(r)]$. Here, $f$ is typically a function of a variable number of arguments, permutation-invariant and monotonic in each argument such as the maximum function. They show this can be achieved directly by using gradients for any baseline $b$ independent of $\tau_i$ such as the leave-one-out baseline $b = f(r_{-i}) := f(r_1, \ldots, \hat{r}_i, \ldots, r_n)$ which omits $r_i$, giving

$$g = \mathbb{E}_{\tau_i \sim \pi_\theta}\left[\sum_i \left(f(r) - f(r_{-i})\right) \nabla_\theta \log \pi_\theta(\tau_i)\right].$$

In the case of hierarchical inference where the solutions are recombined before evaluation, we have a large number of correlated rewards and optimize for large pass@$k$. Unlike the original paper (Tang et al., 2025), we can't take $f$ as the maximum over all samples since it would saturate and lead to extremely sparse advantages: a positive advantage would only be observed if $\tau_i$ is the *only* solution among $k$ attempts that solves the task (see Figure 12 in the Appendix). To adapt multi-sample objective training to this use case, we propose to use soft objective functions that interpolate between the hard maximum (pass@k training) and the average (standard training). We opt for the $\text{logmeanexp}_\beta$ function defined by

$$\text{logmeanexp}_\beta(r_1, \ldots, r_n) = \beta \log\left(\frac{1}{n}\sum_{i=1}^{n} e^{r_i/\beta}\right). \tag{1}$$

This function has several theoretical appeals:

- It is a smooth interpolation between the average $(\beta \to \infty)$ and the maximum $(\beta \to 0)$.

- For a well-chosen value of $\beta$, it closely approximates the average between pass@1 + pass@10 + pass@100 + pass@1000. Such objectives strike a balance between favoring exploration and diversity (due to pass@1000 components) and re-ranking to favor successful solution attempts for exploiting the current knowledge (due to pass@1 components). See Appendix 11 for details and Figure 3.

- It represents a soft form of optimism over the other actions. Recall that Deep-Q-Learning (Mnih et al., 2013) uses $R_t + \max_a Q(s_{t+1}, a)$ as the Q-value target for $Q(s_t, a_t)$ if the action led to the new state $s_{t+1}$. This target can be seen as an approximation of $Q^*(s_t, a_t)$ for the $Q^*$ the state-action values of the optimal policy $Q^*$. In the language of soft reinforcement learning (Cohen et al., 2025), $\mathcal{V}_\beta = \text{logmeanexp}_\beta(r_1, \ldots, r_n)$ can be seen as the $\beta$-soft value function of selecting one out of $n$ rollouts with uniform prior $\pi_0$.

In Figure 3, we show the nonlinear reward transformation induced by $\text{logmeanexp}_\beta$ for several values of $\beta$, alongside several other such *utility functions* for comparison. See Appendix 11 for details on induced utility functions.

Concretely, in the setup of hierarchical inference, we apply this form of reward aggregation as follows. We generate $d$ decompositions $D_i$ of size $n_i$, $1 \le i \le d$, respectively and $k$ implementations for each function in each $D_i$. For each decomposition, we evaluate $m \le k^{\max(n_1, \ldots, n_d)}$ combinations. Write $r$ for all $dm$ rewards obtained in this way, $r_{ij}$ for a single reward, $i \le d, j \le m$ and $A(r_{ij})$ for the set of actions involved in $r_{ij}$.

DecompRL is a cooperative framework of two policies: decomposition $\pi_\theta(D)$ and implementation $\pi_\theta(I \mid D)$. For the decomposition policy, we use the policy gradient

$$g = \mathbb{E}_{r_{ij}} \left[ \sum_{i=1}^n \left( \text{logmeanexp}_\beta(r) - \text{logmeanexp}_\beta(\{r_{i'j} \mid i' \neq i\}) \right) \nabla_\theta \log \pi_\theta(D_i) \right]. \tag{2}$$

For the implementation policy, assume $d = 1$ and write $r_j$ for $r_{1,j}$. Then we use the policy gradient

$$g = \mathbb{E}_{r_j} \left[ \sum_{l \le n, j_l \le k} \left( \text{logmeanexp}_\beta(r) - \text{logmeanexp}_\beta(\{r_j \mid I_l^{j_l} \notin A(r_j)\}) \right) \nabla_\theta \log \pi_\theta(I_l^{j_l} \mid D) \right].$$

In other words, the advantage is computed from the multi-sample objective using all observed rewards and baselined by the multi-sample objective computed from all rewards that an action did not participate in. Because, by this construction, the baseline is independent of the action, it does not bias the gradient estimate. The leave-one-out baseline reduces the overall gradient variance (see Appendix 12 for the computation adapted to multi-sample objectives).

## 3 Experimental setup

### 3.1 Overview

**Datasets and models:** We perform experiments with three different large language models: Qwen 2.5 7B (Qwen et al., 2025), Llama 3.1 8B Instruct (AI @ Meta, 2024) and the Code World Model 32B (FAIR CodeGen team et al., 2025). We train all models with online reinforcement learning on a mix of $15,000$ competitive programming problems including the CodeContest (Li et al., 2022) and TACO Li et al. (2023) training sets. We evaluate on the DeepMind Code Contest validation set (Li et al., 2022) (117 problems) and LiveCodeBench range 2024/08/01 to 2025/02/01 (279 problems) (Jain et al., 2024a). All evaluations are done with temperature 1.0 and top-p 1.0 to promote answer diversity.

**Algorithms:** Similar to Shao et al. (2024); DeepSeek-AI (2025), we optimize policy gradients using Proximal Policy Optimization (PPO) (Schulman et al., 2017) without a critic model using the advantage described in section 2.3. We use an asynchronous distributed RL framework (Gehring et al., 2025; Noukhovitch et al., 2025) where we divide 80 H100s into a set of workers (producing samples) and trainers (updating the current policy). We use 72 workers and 8 trainers. All code generations are evaluated on an external CPU cluster.

**Baselines:** We compare with Group Relative Policy Optimization (GRPO) and algorithms optimized for high pass@$k$ performance specifically: pass@$k$ training (Chen et al., 2025) (set $k = 8$), Soft Policy Optimization (SPO Cohen et al. (2025)) and our own logmeanexp leave-one-out objective (see Equation 2) with $\beta = 0.3$. All methods use 16 samples per prompt. We also include a logmeanexp objective with 48 samples per prompt and $\beta = 0.1$ to mimic the number of forwards per decomposition present in DecompRL training (at most 6 functions and 8 implementations per problem). See hyperparameter and baseline tuning details in Appendix 14 and 15.

### 3.2 DecompRL as Multi-Agent Reinforcement Learning

DecompRL is a form of cooperative multi-agent RL where a decomposition and an implementation policy maximize the number of correct code combinations. Multi-agent reinforcement learning poses challenges not present in single-agent reinforcement learning. Agent policies are interdependent and non-stationary, which raises the questions of how to perform *credit assignment* after jointly observed rewards and how to optimize

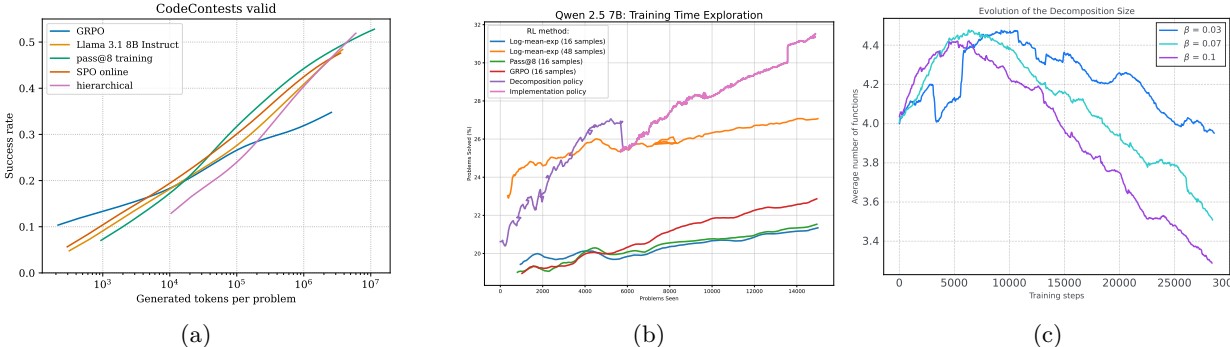

(a)  (b)  (c)

Figure 4: **Diversity from recombined samples. (a)** Scaling inference-compute on CodeContests valid: Standard reinforcement learning for code (GRPO) increases pass@1 while degrading pass@k for large $k$. Our method of hierarchical inference is competitive with scaling inference on the instruct model as well as with diversity-focused reinforcement learning methods pass@8-training and SPO when evaluating only up to 4096 combinations per decomposition. **(b)** Training the implementation and decomposition policies both help increase the solve rate per problem. **(c)** The decomposition policy learns to use less functions during training. We apply a logmeanexp normalization where the $\beta$ can be tuned between mean ($\beta \to \infty$) and max ($\beta \to 0$).

toward a social optimum (in the cooperative case) (Albrecht et al., 2024). To address the moving target problem, we use sequential training and a counterfactual action value estimation (Foerster et al., 2024) with our leave-one-out baseline. Similar to Expectation-Maximization algorithms Rosipal & Girolami (2001), our training has two stages. The decomposition and implementation policies are represented by two different copies of our model. First, we train the decomposition policy for $30k$ steps keeping the implementation policy fixed. Second we train the implementation policy also for $30k$ steps keeping the decomposition fixed. For each problem, we use 8 decomposition samples. For a decomposition with $n$ functions ($n_{max} = 6$), we implement each 8 times yielding $8^n$ code combinations. We randomly sample 512 to evaluate out of $8^n$ to balance exploration between breadth-first (exploring new problems) and depth-first search (scaling evaluations per problem). Although we have two copies of the model only one is trained at a time so this has similar cost as the reference policy used in traditional RL algorithms.

## 4 Results

DecompRL produces large numbers of diverse samples that leads to diverse solutions (Section 4.1), learns modularity (Section 4.2) and solves new problems (Section 4.3)

Table 1: **DecompRL makes the most of high token compute.** We compare pass@tokens: given a set token budgets what is the solve rate across different online RL training methods on LiveCodeBench using Qwen 2.5 7B.

| Token budget | GRPO | instruct | lme16 | lme48 | pass@8 | DecompRL (ours) |
|---|---|---|---|---|---|---|
| 1,000 tokens | 0.18 | 0.06 | 0.19 | 0.12 | 0.15 | 0.18 |
| 5,000 tokens | 0.29 | 0.16 | 0.28 | 0.26 | 0.27 | 0.18 |
| 10,000 tokens | 0.32 | 0.21 | 0.31 | 0.30 | 0.31 | 0.25 |
| 50,000 tokens | 0.38 | 0.30 | 0.38 | 0.36 | 0.39 | **0.40** |
| 100,000 tokens | 0.40 | 0.33 | 0.40 | 0.39 | 0.41 | **0.44** |
| 500,000 tokens | 0.44 | 0.39 | 0.46 | 0.44 | 0.46 | **0.48** |

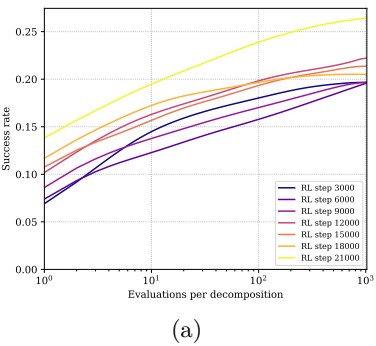 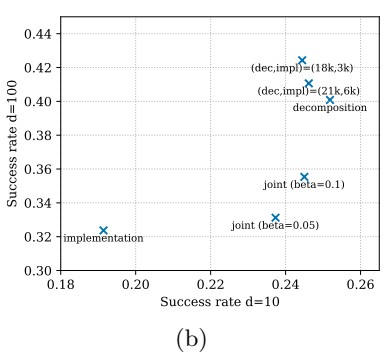 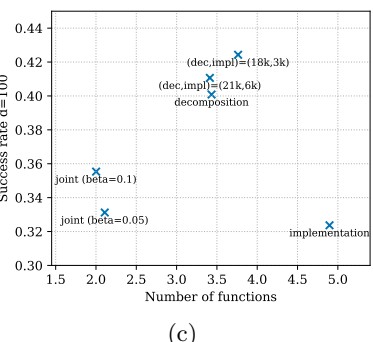

(a)             (b)             (c)

Figure 5: **Recombination is critical for benchmark performance.** A series of analysis done with LLama 3.1 8B Instruct. **(a)** Training a decomposition and implementation policy helps maximize success rate per re-combined code evaluation despite a fixed inference budget. **(b)** Let $d$ the number of decompositions per problem, success rate at $d = 10$ attempts does not predict success rate at $d = 100$ attempts, **(c)** whereas the size of decompositions does. Best performing models for large attempt numbers produce bigger decompositions on average.

### 4.1 DecompRL leads to diverse solutions

A decomposition with $n$ functions and $k$ implementations per function costs $nk + 1$ model forwards and yields $k^n$ (correlated) code combinations for evaluation. For hierarchical inference to make a difference, we need the final solutions to have enough diversity to offset their generation cost compared to $nk + 1$ independent whole-code samples. With the Llama 3.1 8B and Qwen 2.5 7B, we show training the decomposition policy increases success rates per decomposition and number of evaluations.

In Figure 5a, we investigate the diversity of recombined programs by fixing a maximum number $m$ of code evaluations per decomposition and observing the relationship between $m$ and the overall success rate of a DecompRL model. It can be seen that the success rate from less than 50 model forwards per problem shows no sign of plateauing even when scaling $m$ up to 1000 evaluations. Crucially, this remains true throughout reinforcement learning training as shown in Figure 4b where the solve rate continues to increase given a fixed evaluation budget of $m = 512$ code samples.

### 4.2 Recombination matters

The benefit of recombining partial solutions is most pronounced when the decomposition size $n$ is large where $n$ is the number of functions picked by the model. In our experiments, we see that models with similar success rates using 10 decompositions behave very differently when inference is scaled to 100 decompositions. As we scale the inference budget per problem, models that create larger decompositions improve faster than those with smaller ones. The success rate with 10 decomposition attempts does not predict the success rate with 100 attempts, but the size of the decomposition does (see Figures 5b and 5c).

### 4.3 Solving new problems

DecompRL through recombination and diverse evaluations solves new problems that are harder (Figure 2) and only solvable at high inference budgets (Figure 3 and Table 1). Using different models with DecompRL training, we beat the best existing methods for reinforcement learning when using high inference budgets (up to 3 million tokens per problem). We evaluate $m = 4096$ combinations per decomposition, which is roughly 80 times more than $nk + 1$ language model calls would have produced with standard whole-code generation. On the other hand, $m = 4096$ is small compared to the theoretical maximum $k^{n_{\max}} = 8^6$ of combinations that hierarchical inference can produce. Additional CPU execution resources could further increase the appeal of the method according to the scaling behavior in Figure 4. DecompRL can help overcome the limits of our SFT checkpoint and find solutions not available with regular inference. We see it as a useful way to gather offline data to improve the starting policy.

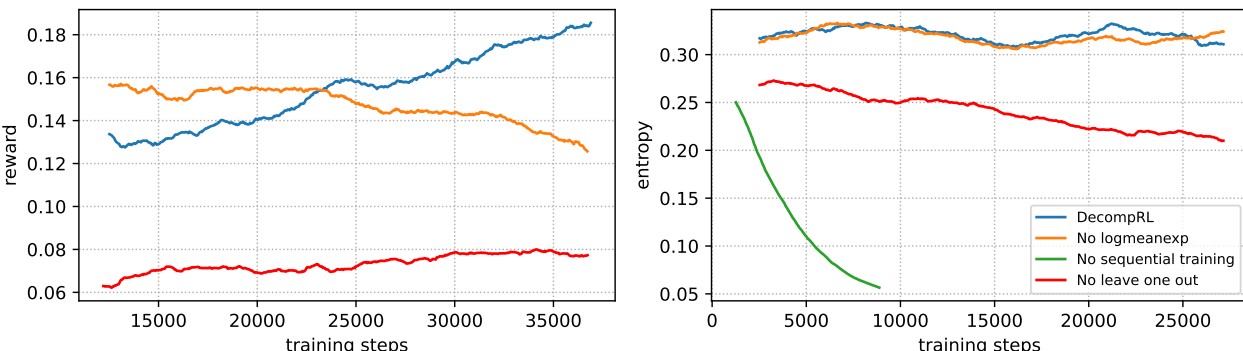

Figure 6: **Ablation study on the DecompRL objective,** removing logmeanexp for a mean aggregation over rewards using a softmax, removing sequential training for joint training of both policies, and the leave one out baseline for value estimation in the advantage all lead to entropy or reward collapse.

## 5 Limits

**Format tax.** DecompRL with Llama 3.1 8B Instruct model degrades performance from 26.6% to 14.5% on the CodeContest validation with the same token budget (10 attempts with DecompRL, corresponding to pass@316 for standard inference). On the "easy" split of LiveCodeBench, DecompRL performs worst than the starting policy using Qwen 2.5 7B (see Figure 2). Looking at the CodeContest training set as an example of competitive programming data, we notice decomposition and function generation are off-policy tasks for the model. Only 65 % of the provided solutions in the CodeConsests training set contain functions. We expect data in the training stage of the Llama 3.1 8B Instruct model to follow a similar distribution and potentially not encourage modular coding. The model has also been heavily fine-tuned through RLHF (Christiano et al., 2023), human annotations and synthetic data for the specific task of generating full code answers when prompted with problems in the standard inference setup. DecompRL unlike other reinforcement learning algorithms has to overcome this off-polyciness during training.

**Size collapse.** Hierarchical inference comes with the cost of reduced precision at the single sample level. Decomposition training leads to a decreasing decomposition size (see Figure 4c). This can allow for more relevant functions but in the limit if we have a single function, hierarchical inference becomes equivalent to classical whole-code generation. We see this as a form of reward hacking during training which can be delayed by using a lower $\beta$ in the logmeanexp objective and alternating between implementation and decomposition training. Simpler approaches such as bonuses based on the decomposition size led to reward hacking.

**Sampling time.** During training, DecompRL scales the number of evaluations per problem to up to 512 in our experiments compared to the default 16. This allows for more exploration during training (see Figure 5a) but can also lead to: slower worker GPUs and introduce off-polyciness in online RL. We see DecompRL as a policy distillation method for gathering new solutions on the training set not available with regular RL. If we have a good enough starting policy using regular RL remains more efficient as a post training strategy.

## 6 Ablations

In standard RL with LLMs, one action leads to one reward and with GRPO we compute the advantage over a group $G$ of rewards where $G << 100$ due to inference limitations. In DecompRL, we instead have two policies performing $k \times n$ actions which lead to $k^n$ rewards. The advantage of each action is therefore estimated by many indirect and correlated rewards. In order to prevent over-estimating certain actions, we introduce in equation 2: 1) a leave one out instead of mean baseline, 2) transforming the rewards per action using a logmeanexp function (Equation 1) 3) training sequentially a set of two policies. Figure 6 shows how each component prevents instabilities. We run ablations on the Code World Model 32B.

**Ablation on logmeanexp.** We compare our max interpolation (logmeanexp, $\beta = 0.03$) with a mean interpolation (softmax, $\beta = 0.5$). As we have many noisy rewards optimizing towards the mean eventually leads to reward collapse. Lower $\beta$ in logmeanexp also leads to faster decomposition size decay (see Figure 4c).

**Ablation on sequential training.** Instead of alternating between training the implementation and decomposition policies, we study two other strategies: training only the implementation policy, and jointly training both policies (meaning no frozen model). Without clear credit assignment and with an over representation of implementation samples, we overfit the implementation policy and underfit the decomposition policy. Joint training leads to entropy collapse (Figure 6) whereas sequential training allows us to keep scaling the number of functions with increased gains in performance (Figure 5).

**Ablation on leave one out baseline.** Removing the leave one out baseline in the advantage calculation for both policies leads to much lower rewards and earlier plateau.

## 7 Related works

**Scaling inference methods.** Intermediate steps to help models generate correct code solutions has been studied at the "plan", "algorithm" and "function" level (Khot et al., 2023; Zhou et al., 2024; 2023; Jain et al., 2024b; Wang et al., 2025). We adapt the decomposition and function approach from Zelikman et al. (2023). To promote sampling diversity, optimizing over code generation trajectories has been studied mainly with repeated sampling (Li et al., 2022), self-refinement chains with execution feedback (Shinn et al., 2023; Madaan et al., 2023), and tree search using multiple children solutions per chain Feng et al. (2024); Yao et al. (2023); Light et al. (2025). Tree based generation systematically organizes and potentially directs linear generation, but generally does not create a combinatorial increase in solutions to evaluate (each evaluated leaf has to have been generated).

**Recombination of partial solutions.** Methods that decompose a problem into separate parts which can be solved independently and then recombined have been studied for solving mathematical and coding tasks. In mathematical theorem proving, tree search in formal theorem proving systems decomposes the problem into sub-goals that can be solved and checked independently by the verifier(Lample et al., 2022; Polu et al., 2022). In code generation and symbolic regression, library learning creates solutions to sub-problems for later reuse (Ellis et al., 2023; Gauthier et al., 2023; Grayeli et al., 2024). These methods study re-combination at the inference level with a fixed policy for determining blocks whereas we propose reinforcement learning to improve the policy.

**Increasing code modularity** Programs can be analyzed hierarchically (Shi et al., 2024) or generated hierarchically (Zelikman et al., 2023). Via clustering in the embedding space (CodeChain (Le et al., 2023)), prompting different agents MapCoder (Islam et al., 2024)) and iterating on trees of functions based on shared consensus (Divide and Conquer (Chen et al., 2024)), inference methods have enforced modularity in code generation trajectories. We go beyond inference structure through dedicated RL training which could be combined with these frameworks.

**Self-play reinforcement learning.** In formal mathematical proving, a *conjecturer* model asserts statements for a prover to prove, either as a self-guided exploration task with intrinsic motivation (Poesia et al., 2024) or as a live augmentation technique during a standard reinforcement learning run for theorem proving (Dong & Ma, 2025). In the domain of code, self-play involves a programming puzzle generator and a model that solves the generated tasks (Haluptzok et al., 2022; Teodorescu et al., 2023). In DecompRL, the decomposition model can be viewed as a self-play teacher that is conditioned on the underlying programming problem to be solved.

**Expert Iteration.** Expert iteration (Anthony et al., 2017) divides the problem into an "expert" (a slow tree search algorithm doing the heavy exploration) and an "apprentice" (a fast neural network). As DecompRL is optimized for harder problems and high pass@$k$, we can view it as the "expert" policy gathering novel solutions to train a starting RL policy offline (the "apprentice"). Online expert iteration methods such as STaR (Zelikman et al., 2022), Quiet-STaR (Zelikman et al., 2024), ReST (Gulcehre et al., 2023), and ReST-MCTS Zhang et al. (2024) use the same model as expert and apprentice to iteratively bootstrap its

reasoning capabilities using sucessful trajectories. Similar our hierarchical inference process, ReST-MCTS goes beyond regular inference using tree search to find solutions normally missed by a weak starting policy.

## 8 Conclusion

Solving programming problems by decomposing them into simpler subproblems and recombining them into full solutions allows to scale inference-time evaluation budgets with a limited number of language model calls. However, off-the-shelf instruction finetuned models do not fare well at this task. We introduce DecompRL, a reinforcement learning technique that directly trains models for hierarchical inference, greatly enhancing their performance on this task. This method encourages diversity by directly optimizing for a mixture of pass@k objectives with a novel logmeanexp advantage function. The resulting models use recombination to solve problems unreachable with regular RL training. With evaluation up to pass@1k or $10^7$ generated tokens per problem, we show with different models (Llama 3.1 8B Instruct, Qwen 2.5 7B) that the success rate of DecompRL continues to scale whereas other methods begin saturating. With larger CPU resources, we envision the potential for significant advancements when scaling such systems to millions of evaluated answers. Moreover, the modularity of the resulting generation pipeline offers avenues for interpretability insights and an axis of control which is not available in other methods of scaling inference compute, such as in repeated sampling or scaled chain-of-thought.

Looking forward, many scientific and engineering tasks are more easily evaluated than solved, presenting opportunities for innovative solutions. For writing entire codebases that solve specific problems, models could achieve acceptable pass rates at scale (e.g., pass@1M), even if their initial performance (pass@1) is poor. Our findings suggest that DecompRL is a high performing search policy to solve new and harder problems and could become an ingredient of any training pipeline where recombination is possible and evaluation is cheap while generation is expensive.

More generally, our findings shed light onto the hard *exploration question* in LLM post-training (Cui et al., 2025): How can models discover new behaviors that allow solving previously unsolved tasks without the need for human annotations, and more efficiently than via finetuning on large-scale rejection sampled datasets (e.g. in formal math Polu et al., 2022; Lin et al., 2025)? While DecompRL is specific to code generation, it hints at a possible future where *exploration policies* are trained specifically for creating high-quality data with behaviors that the final, *exploiting policy* trained with a standard pass@1 objective could not have found by itself over the course of its training.

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

# Appendix

# 9 Proofs

## 9.1 Variance Reduction

In this section, we give the proof for the theorem on variance reduction by evaluating recombinations. First, we give an accessible direct proof in the case of two components that exhibits the key idea, then we extend to an arbitrary amount of components.

**Theorem 9.1.** *Let $n \geq 1$, $X_1, \ldots, X_n, Y_1, \ldots, Y_n$ be independent random variables such that all $X_i$ are identically distributed and all $Y_j$ are identically distributed. Let $f(X, Y)$ be a scalar function and assume $\mathbb{E} f(X_1, Y_1)^2 < \infty$. Then*

$$\operatorname{Var} \frac{1}{n^2} \sum_{i,j} f(X_i, Y_j) \leq \operatorname{Var} \frac{1}{n} \sum_i f(X_i, Y_i) = \frac{1}{n} \operatorname{Var} f(X_1, Y_1).$$

*Proof.* The equality follows from independence of $(X_i, Y_i)$ and $(X_j, Y_j)$ for $i \neq j$. For the proof of the inequality, write $f_X = E[f(X, Y) \mid X]$, $f_Y = E[f(X, Y) \mid Y]$, $\mu = \mathbb{E}[f(X, Y)]$ and consider the Hoeffding decomposition

$$\operatorname{Var} f(X, Y) = \operatorname{Var} f_X + \operatorname{Var} f_Y + \operatorname{Var} \left( f(X, Y) - f_X - f_Y + \mu \right),$$

which is proved by expanding all terms and applying the law of total expectation. In particular, by non-negativity of the variance, we have

$$\operatorname{Var} f_X + \operatorname{Var} f_Y \le \operatorname{Var} f(X, Y).$$

To apply this result, write

$$\operatorname{Var} \frac{1}{n^2} \sum_{i,j} f(X_i, Y_j)$$

$$= \frac{1}{n^4} \sum_{i,j,k,l} \operatorname{Cov}(f(X_i, Y_j), f(X_k, Y_l))$$

$$= \frac{1}{n^4} \sum_{i,j,k,l, i=k \text{ or } j=l} \operatorname{Cov}(f(X_i, Y_j), f(X_k, Y_l))$$

$$= \frac{1}{n^4} \left( \sum_{i,j} \operatorname{Var} f(X_i, Y_j) + \sum_{i,j,l,j \ne l} \operatorname{Cov}(f(X_i, Y_j), f(X_i, Y_l)) + \sum_{i,j,k,k \ne i} \operatorname{Cov}(f(X_i, Y_j), f(X_k, Y_j)) \right)$$

$$= \frac{1}{n^2} \operatorname{Var} f(X, Y) + \frac{n-1}{n^2} \operatorname{Var} f_X + \frac{n-1}{n^2} \operatorname{Var} f_Y$$

$$\le \frac{1}{n^2} \operatorname{Var} f(X, Y) + \frac{n-1}{n^2} \operatorname{Var} f(X, Y)$$

$$= \frac{1}{n} \operatorname{Var} f(X, Y).$$

Here, we used the fact that $f(X_i, Y_j)$ and $f(X_k, Y_l)$ are independent if $i \ne k$ and $j \ne l$ and made sure not to count the pair of index pairs $(i, j), (i, j)$ twice. $\qquad\square$

The generalization to $m$ variables is given in the following theorem. The proof proceeds in the same way but requires the general Hoeffding decomposition involving marginals over all subsets of variables (Vaart, 1998, Chapter 11.4).

**Theorem 9.2.** *Let $n, m \ge 1$, $X_i^j$, $i \le m$, $j \le n$ be independent random variables such that $X_i^j, X_i^k$ are identically distributed for all $i, j, k$. Let $f(X_1, \ldots, X_m)$ be a scalar function and assume $\mathbb{E} f(X_1^1, \ldots, X_m^1)^2 < \infty$. Then*

$$\operatorname{Var} \frac{1}{n^m} \sum_{j_i} f(X_1^{j_1}, \ldots, X_m^{j_m}) \le \operatorname{Var} \frac{1}{n} \sum_j f(X_1^j, \ldots, X_m^j) = \frac{1}{n} \operatorname{Var} f(X_1^1, \ldots, X_m^1).$$

*Proof.* Write $f = f(X_1, \ldots, X_m)$ for short and consider the Hoeffding decomposition defined recursively by

$$f_\varnothing = \mathbb{E}[f],$$

and for $A \subseteq \{1, \ldots, m\}$,

$$f_A = E[f \mid X_i, i \in A] - \sum_{B \subsetneq A} f_B.$$

This decomposition is orthogonal (i.e. $\mathbb{E}[f_A f_B] = 0$ if $A \ne B$) (Vaart, 1998, Lemma 11.11), and we have

$$f = \sum_{S \subseteq \{1, \ldots m\}} f_S$$

and thus

$$\operatorname{Var} f = \sum_S \operatorname{Var} f_S.$$

We now proceed with the proof by writing the variance of the sum into covariance terms and counting contributions like in the case of $m = 2$ above. Write $j = (j_1, \ldots, j_m)$ and $k = (k_1, \ldots, k_m)$ for two multi-indices, $j \cap k = \{i \mid j_i = k_i\}$ for the positions where they agree and $f^j$ for $f(X_1^{j_1}, \ldots, X_m^{j_m})$. We have

$$\operatorname{Var} \frac{1}{n^m} \sum_j f^j$$

$$= \frac{1}{n^{2m}} \sum_{j,k} \operatorname{Cov}(f^j, f^k)$$

$$= \frac{1}{n^{2m}} \sum_{j,k} \operatorname{Cov}(\sum_S f_S^j, \sum_T f_T^k)$$

$$= \frac{1}{n^{2m}} \sum_{j,k} \sum_{S \subseteq j \cap k} \operatorname{Cov}(f_S^j, f_S^k)$$

$$= \frac{1}{n^{2m}} \sum_{j,k} \sum_{S \subseteq j \cap k} \operatorname{Var} f_S$$

by orthogonality and the fact that $f_S^j$ and $f_S^k$ depend on the same random variables.

What is the coefficient of the term $\operatorname{Var} f_S$ in this sum? Since the indices must agree at the positions in $S$, there are $n$ options for each $j_i = k_i$ for $i \in S$, giving $n^{|S|}$ combinations. At the other indices $i \notin S$, the indices $j_i$ and $k_i$ can take any values ($j_i = k_i$ is possible because $S \subseteq j \cap k$, not $S = j \cap k$), giving another $n^{2m-2|S|}$ combinations and a total of $n^{2m-|S|}$. Therefore, we get

$$\operatorname{Var} \frac{1}{n^m} \sum_j f^j = \frac{1}{n^{2m}} \sum_{j,k} \sum_{S \subseteq j \cap k} \operatorname{Var} f_S = \frac{1}{n^{2m}} \sum_S n^{2m-|S|} \operatorname{Var} f_S = \sum_S \frac{1}{n^{|S|}} \operatorname{Var} f_S$$

and using that $\operatorname{Var} f_\varnothing = \operatorname{Var} \mathbb{E}[f] = 0$ and $|S| \geq 1$ for $S \neq \varnothing$,

$$= \sum_{S \neq \varnothing} \frac{1}{n^{|S|}} \operatorname{Var} f_S = \sum_{S \neq \varnothing} \frac{1}{n^{|S|}} \operatorname{Var} f_S \leq \sum_{S \neq \varnothing} \frac{1}{n} \operatorname{Var} f_S = \frac{1}{n} \operatorname{Var} f$$

according to the variance decomposition stated above. □

The bound in the theorem is tight, as evidenced by functions $f$ which are linear in scalar random variables $X_i$. In the other extreme, if $f(X_1, \ldots, X_n) = \prod_i X_i$ and $\mathbb{E}[X_i] = 0$ for all $i$, then all non-diagonal covariance terms disappear and the left-hand side is $\frac{1}{n^{2m-1}} \operatorname{Var} f$ while the right-hand side is $\frac{1}{n} \operatorname{Var} f$.

Let $X_1, \ldots X_n$ and $Y_1, \ldots, Y_n$ be independent random variables such that $X_i$ identically distributed and $Y_i$ identically distributed. Let $f$ be a deterministic function such that $Z_i = f(X_i, Y_i)$ has finite mean $\mu$ and finite variance $\sigma^2$.

According to the central limit theorem, with $S_n = \frac{Z_1 + \ldots + Z_n}{n}$ we have

$$\sqrt{n}(S_n - \mu) \to (0, \sigma^2)$$

in distribution.

Now consider the recombining estimator

$$T_n = \frac{1}{n^2} \sum_{i,j \leq n} f(X_i, Y_j).$$

Clearly, $\mathbb{E} T_n = \mu$. In theorem 9.2 we established

$$\operatorname{Var} T_n \leq \operatorname{Var} S_n = \frac{1}{n} \sigma^2$$

using the Hoeffding decomposition of $f(X_i, Y_j)$.

More precisely, setting $f_X = \mathbb{E}[f(X,Y) \mid X] - \mu$, $f_Y = \mathbb{E}[f(X,Y) \mid Y] - \mu$, $f_{XY} = f(X,Y) - f_X - f_Y - \mu$ and $f'(X,Y) = \mu + f_X + f_Y + \frac{1}{\sqrt{n}} f_{XY}$, we derived (implicitly in the proof, and noting the change of notation for $f_X$, $f_Y$)

$$\operatorname{Var} T_n = \frac{1}{n} \operatorname{Var} f_X + \frac{1}{n} \operatorname{Var} f_Y + \frac{1}{n^2} \operatorname{Var} f_{XY} = \frac{1}{n} \operatorname{Var} f'.$$

Now set $S'_n = \frac{f'(X_1, Y_1) + \ldots + f'(X_n, Y_n)}{n}$, observe that $\mathbb{E} f' = \mu$ and write

$$\sigma'^2 = \operatorname{Var} f' = \operatorname{Var} f_X + \operatorname{Var} f_Y + \frac{1}{n} \operatorname{Var} f_{XY} \leq \sigma^2.$$

According to the central limit theorem,

$$\sqrt{n}(S'_n - \mu) \to (0, \sigma'^2)$$

We will now prove that [1]

$$\sqrt{n}\|T_n - S'_n\|_2 \to 0$$

with $n \to \infty$. Indeed,

$$n\|T_n - S'_n\|_2^2 = \frac{1}{n^3} \sum_{i,j,k,l} \left( f(X_i, Y_j) - f'(X_i, Y_i) \right) \left( f(X_k, Y_l) - f'(X_k, Y_k) \right)$$

Subtracting $\mu$ from all $f$- and $f'$-terms, proceeding as in the proof of Prop. B.1 and observing that $\operatorname{Cov}(f(X_i, Y_j), f'(X_i, Y_i)) = \operatorname{Var} f_X$, $\operatorname{Cov}(f(X_i, Y_j), f'(X_j, Y_j)) = \operatorname{Var} f_Y$, $\operatorname{Cov}(f(X_i, Y_i), f'(X_i, Y_i)) = \operatorname{Var} f'$, we get

$$n\|T_n - S'_n\|_2^2 = \frac{1}{n^3} \left( n^2 \operatorname{Var} f + n^3 \operatorname{Var} f' - n^2(n-1) \operatorname{Var} f_X - n^2(n-1) \operatorname{Var} f_Y \right)$$

$$= \frac{1}{n^3} \left( n^2 \operatorname{Var} f + n^3 (\operatorname{Var} f_X + \operatorname{Var} f_Y + \frac{1}{n} \operatorname{Var} f_{XY}) - n^2(n-1) \operatorname{Var} f_X - n^2(n-1) \operatorname{Var} f_Y \right),$$

so all $n^3$ terms within the brackets cancel and

$$n\|T_n - S'_n\|_2^2 \to 0$$

with $n \to \infty$ as claimed.

Since $\sqrt{n}(T_n - \mu) \to \sqrt{n}(S'_n - \mu)$ in $L_2$ and $\sqrt{n}(S'_n - \mu) \to (0, \sigma'^2)$ in distribution, $\sqrt{n}(T_n - \mu) \to (0, \sigma'^2)$ in distribution, satisfying a CLT.

We now state the Delta Method theorem [2].

**Theorem 9.3.** *Let $S_n$ be a sequence of random variables that satisfies $\sqrt{n}(S_n - \mu) \to (0, \sigma^2)$ in distribution. For a given function $u$, suppose that $u'(\mu)$ exists and is not 0.*

*Then,*

$$\sqrt{n}(u(S_n) - u(\mu)) \to (0, \sigma^2 u'(\mu)^2)$$

*in distribution.*

We have seen that in the setup of our paper with $u$ a concave utility function implicitly defined by a reward aggregation both the recombination estimator $T_n$ and the standard (diagonal) estimator $S_n$ satisfy a CLT with variances $\sigma'^2$ and $\sigma^2$, respectively. Applying the Delta Method theorem, we can conclude that $u(T_n)$ and $u(S_n)$ likewise satisfy a CLT but with different variances $\sigma'^2 u'(\mu)$ and $\sigma^2 u'(\mu)$, respectively. Since $\sigma'^2 = \sigma^2 - \frac{n-1}{n} \operatorname{Var} f_{XY}$, the recombination estimator is an improvement over the standard estimator, not just for $u = \operatorname{id}$, but also in the presence of nonlinear utility functions (reward aggregation functions) $u$ in the $n \to \infty$ limit.

---

[1] similar to https://math.stackexchange.com/questions/2760951/central-limit-theorem-for-non-degenerate-u-statistics
[2] https://www.stat.rice.edu/~dobelman/notes_papers/math/TaylorAppDeltaMethod.pdf

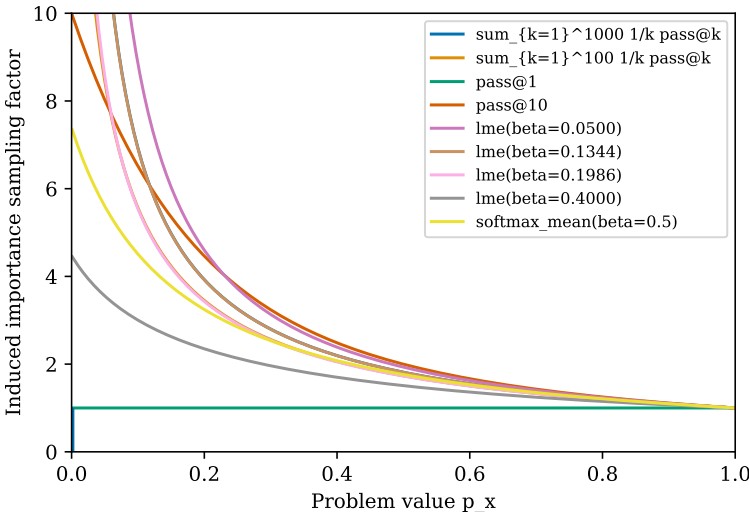

Figure 7: **Induced importance sampling weights of explorative utility functions.** The functions are the same ones as in Figure 3.

## 10 Information-Theoretic View on Inference Scaling

Consider a dataset $D$ of problems $x \in D$, an evaluation function $v(x, y) \in \{0, 1\}$, a probabilistic model $\pi$ and consider the pass rate or coverage function

$$c(k) = \mathbb{E}_{x \sim D, y_1, \ldots, y_k \sim \pi(x)}[\max_i v(x, y_i)].$$

Empirical investigations (Brown et al., 2024; Hughes et al., 2024; OpenAI, 2023)frequently report power law scaling of the logarithmic pass rate with respect to the number of samples of the form

$$\log(c) \approx -ak^{-b},$$

where $a, b > 0$.

To analyze this relationship, note that $k$ can be seen as an expected waiting time in the Bernoulli process with repeated samples from $v(x, \pi(x)) \sim \text{Bern}(p)$ where $p = \frac{1}{k}$, and we can consider the information quantity $h = -\log p = \log k$. Under the logarithmic power law model, we know the cumulative distribution function

$$\mathbb{P}(k \leq k_0) = c(k_0) = \exp\left(-ak^{-b}\right)$$

and thus for $h_0 = \log k_0$,

$$\mathbb{P}(h \leq h_0) = \exp\left(-ae^{-bh}\right).$$

Differentiating, we get the probability density function

$$p(h) = ab \exp(-bh - ae^{-bh}) = \frac{1}{\beta} \exp\left(-\frac{h - \mu}{\beta} - e^{-\frac{h-\mu}{\beta}}\right),$$

i.e. a Gumbel distribution with mode parameter $\mu = -\frac{\log a}{b}$ and scale parameter $\beta = \frac{1}{b}$.

Note that the Gumbel distribution arises as an extreme value distribution over exponentially distributed samples, and we can interpret this by imagining several "latent dimensions of problem difficulty" $h_i$, the largest of which determines the overall difficulty of a given problem: $h = \max_i h_i$.

## 11 Reward Aggregation as Utility Functions

Consider as in Section 2.4 $n$ rewards $r_1, \ldots, r_n$. For zero-one rewards $r_i \in \{0, 1\}$, and different choices of a multi-sample objective function $f$, we write $c = \sum_i r_i$ and factor $f(r_1, \ldots, r_n) = u(p)$ via $p = \frac{c}{n}$. Note that $p$ is the mean estimator of $\mathbb{E}[r]$ if $r_i$ are samples from the distribution of $p$.

**Logmeanexp.** We compute

$$f(r_1, \ldots, r_n) = \beta \log \frac{1}{n} \sum_i e^{r_i/\beta} = \beta \log \frac{1}{n} \left( ce^{1/\beta} + (n - c) \right) = \beta \log \left( pe^{1/\beta} + 1 - p \right).$$

**Pass@k with replacement.** In this case, $f(r_1, \ldots, r_n)$ is already defined by

$$f(r_1, \ldots, r_n) = 1 - \left( 1 - \frac{c}{n} \right)^k = 1 - (1 - p)^k.$$

**Softmax-weighted average.** Tang et al. (2025) suggest the multi-sample objective $\sum_i \frac{e^{r_i/\beta}}{\sum_j e^{r_j/\beta}} r_i$. In this case, we obtain

$$f(r_1, \ldots, r_n) = \sum_i \frac{e^{r_i/\beta}}{\sum_j e^{r_j/\beta}} r_i = \frac{ce^{1/\beta}}{ce^{1/\beta} + (n - c)} = \frac{pe^{1/\beta}}{pe^{1/\beta} + (1 - p)} = 1 - \frac{1 - p}{pe^{1/\beta} + 1 - p}.$$

Utility functions are a convenient method to understand the impact of a given choice of multi-sample objective function. We call a function $u : [0, 1] \to [0, 1]$ a *utility function* if it is monotonically increasing. We call a utility function *explorative* if it is concave. Examples of explorative utility functions have been derived above, and are depicted in Figure 3.

For a single problem $x$, we have

$$\mathbb{E}_{r_i \sim \pi(x)}[f(r_1, \ldots, r_n)] = u(\mathbb{E}_{r \sim \pi(x)}[r]),$$

and thus the same holds when taking expectations over $x$. In other words, reinforcement learning with multi-sample objective functions changes the objective to a utility-transformed objective. On a single problem, by monotonicity of $u$, optimal policies for the original objective are optimal for the transformed objective and vice versa. When averaging over a dataset of problems, however, utility functions induce a different allocation of optimization budget. Concretely, let $dx$ be the distribution over problems and $p_x = \mathbb{E}_{r \sim \pi(x)}[r]$. Then

$$\mathbb{E}_x[p_x] = \int_x p_x dx,$$

and

$$\mathbb{E}_x[u(p_x)] = \int_x u(p_x) dx = \int_x p_x \frac{u(p_x)}{p_x} dx,$$

so $\frac{u(p_x)}{p_x} dx$ can be seen as the importance-reweighted distribution of problem difficulties according to $u$. Explorative utility functions are therefore the utility functions that induce a monotonically decreasing reweighting on problem difficulties, as depicted in Figure 7.

## 12 Variance Reduction with Leave-One-Out Baselines

Consider as in Appendix 11 a multi-sample objective function $f$ of zero-one rewards $r_1, \ldots, r_n$ that factors via the mean $p = \frac{1}{n} \sum_i r_i$ via a utility function $u$:

$$f(r_1, \ldots, r_n) = u(p(r_1, \ldots, r_n)).$$

Writing $g_i = \nabla_\theta \log \pi_\theta(a_i)$, we are going to compare the default gradient estimator

$$G_1 = \sum_i f(r_1, \ldots, r_n) g_i = \sum_i u(p) g_i$$

with the baselined one

$$G_2 = \sum_i \left( f(r_1, \ldots, r_n) - f(r_1, \ldots, r_{i-1}, r_{i+1}, \ldots, r_n) \right) g_i = \sum_i \left( u(p) - u(p_{-i}) \right) g_i,$$

where

$$p_{-i} = \frac{1}{n-1} \sum_{j \neq i} r_j.$$

Because $\mathbb{E}[u(p_{-i})g_i] = 0$, we have $\mathbb{E}[G_2] = \mathbb{E}[G_1] = \nabla_\theta \mathbb{E}[f(r_1, \ldots, r_n)]$ by the policy gradient theorem.

We will now compare the covariances of $G_1$ and $G_2$, respectively, assuming $u \in C^2(\mathbb{R}_+, \mathbb{R}_+)$ for convenience and setting $\mu = \mathbb{E}[r_1]$. We have

$$p = p_{-i} + \frac{1}{n}(r_i - p_{-i}) \tag{3}$$

and

$$u(p) = u(p_{-i}) + \frac{1}{n}u'(p_{-i})(r_i - p_{-i}) + \mathcal{O}\left(\frac{1}{n^2}\right),$$

hence

$$\mathrm{Cov}\, G_2 \approx \mathrm{Cov}\left[\sum_i \frac{1}{n}u'(p_{-i})(r_i - p_{-i})g_i\right] \approx \mathrm{Cov}\left[\frac{u'(\mu)}{n}\sum_i (r_i - \mu)g_i\right] = \frac{u'(\mu)^2}{n}\mathrm{Cov}\left[(r_1 - \mu)g_1\right] \in \mathcal{O}\left(\frac{1}{n}\right)$$

by independence of $(r_i - \mu)g_i$ and $(r_j - \mu)g_j$ for $i \neq j$.

For $G_1$, on the other hand, we obtain,

$$\mathrm{Cov}\, G_1 = \mathrm{Cov}\left[\sum_i u(p)g_i\right] \approx \mathrm{Cov}\left[\sum_i u(\mu)g_i\right] = nu(\mu)^2\,\mathrm{Cov}\left[g_1\right] \in \mathcal{O}(n)$$

Since $\mathbb{E}[G_1] = \mathbb{E}[G_2] = \nabla_\theta \mathbb{E}[u(\mu)]$ is independent of $n$ in the limit, we obtain signal-to-noise ratios of

$$\mathrm{SNR}(G_1) = \frac{\|\mathbb{E}[G_1]\|}{\mathrm{Tr}[\mathrm{Cov}\, G_1]} \in \mathcal{O}\left(\frac{1}{n}\right)$$

and

$$\mathrm{SNR}(G_2) = \frac{\|\mathbb{E}[G_2]\|}{\mathrm{Tr}[\mathrm{Cov}\, G_2]} \in \mathcal{O}(n),$$

making $G_1$ inconsistent. The estimator $G_2$ with the leave-one-out baseline, on the other hand, is asymptotically deterministic.

Note that for simplicity, in the above, we considered a leave-one-out baseline that removes one out of $n$ terms from the aggregation. In the case of DecompRL, we remove a fraction of $\frac{1}{d}$ out of $md$ terms in the case of decomposition actions and a fraction of $\frac{1}{k}$ terms out of $m$ terms in the case of implementation actions. Since Equation 3 holds with $n$ replaced by $d$ or $k$, resp., and $r_i$ replaced by the average over the removed terms, the same asymptotic analysis holds.

# 13 Additional Experiments on Multi-Policy Training

We perform experiments at the 8B scale using the LLama 3.1 8B Instruct model AI @ Meta (2024) to derive our multi-policy training recipe described in section 3. We train on the CodeContest training set and evaluate on its validation set. We give additional experiments regarding multi-policy training: benchmark results of combinations of separately trained policies (Figure 10), results on sequential training (Figure 9) and an overall comparison of multi-policy training methods over the course of reinforcement learning runs including alternate training (Figure 8). See Section 3.2 for additional discussion.

Analyzing the policy entropies for decomposition and implementation in separate training runs, we find that entropy decreases substantially over the course of training, hinting at better certainty about good actions (exploitation) but also harming diversity at large sampling scales (exploration). The conditioning of a fixed implementation policy on an entropy-decreasing decomposition policy leads the implementation policy entropy unaffected (Figure 11).

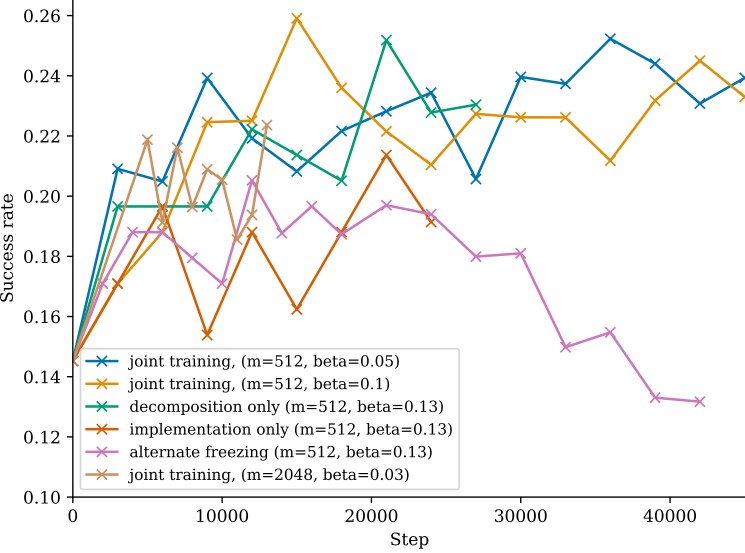

Figure 8: **Decomposition policy is the bottleneck.** Let $m$ the number of evaluations, $\beta$ the logmeanexp coefficient in our advantage normalization, we compare different ablations on the joint training objective: alternate freezing, implementation only, and decomposition only for CodeContest validation set pass@10. Removing training on implementations doesn't hurt model performance suggesting the main bottleneck for RL to overcome is in learning to decompose problems.

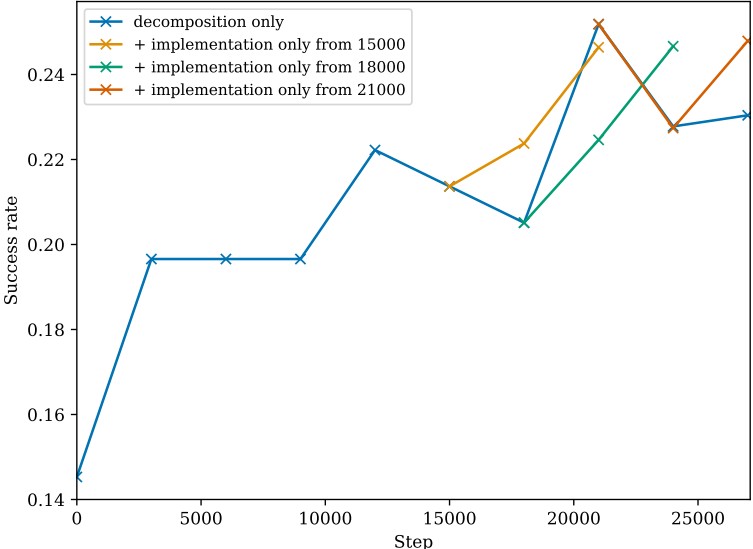

Figure 9: **Sequential training of decomposition and implementation policy.** Shown is a run in which we only train the decomposition policy against a the fixed implementation policy of Llama 3.1 8B Instruct. At 15k, 18k and 21k steps, we continue the training from the checkpoint with training on the implementation policy for 6000 steps. The resulting three checkpoints are all among the best, but within the variance of the decomposition-only evaluation results. Evaluations are conducted on CodeContests valid with 4096 function evaluations per decomposition, $k = 8$ implementations per function and 10 decompositions per problem.

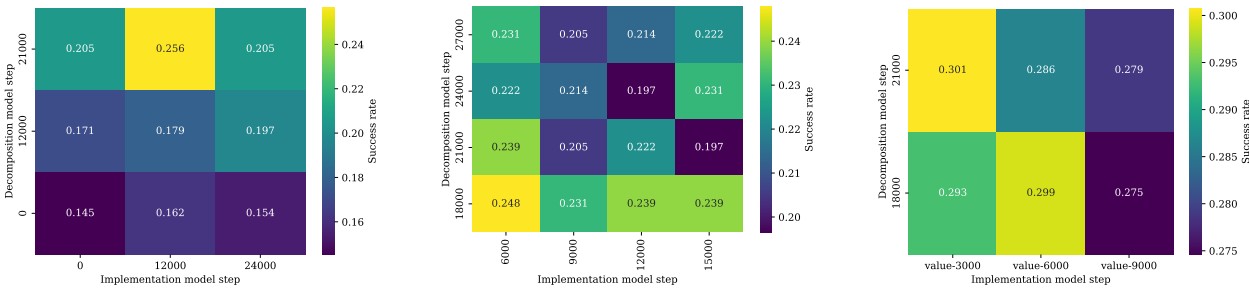

Figure 10: **Performance of different combinations of independently trained decomposition and implementation models.** We train a model on decompositions with implementations from fixed Llama 3.1 8B Instruct, and another model on implementations with decompositions from fixed Llama 3.1 8B Instruct, and evaluate cross combinations between the resulting checkpoints. This off-policy estimation is not guaranteed to work in principle but some combinations appear to work in practice. Note that the best configuration is the decomposition checkpoint with 21k steps coupled with the implementation checkpoint with 3k steps, highlighting the importance of joint optimization. Evaluations are conducted on CodeContests valid with 4096 function evaluations per decomposition, $k = 8$ implementations per function and 10 decompositions per problem.

## 14 Baselines

We compare DecompRL with four reinforcement learning algorithms:

- GRPO (Shao et al., 2024) - the state of the art for optimizing pass@1 performance.
- analytical pass@$k$ (Chen et al., 2025) - the state of the art for optimizing pass@$k$.

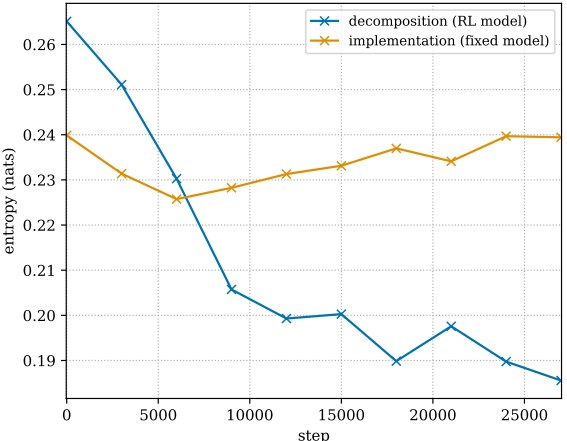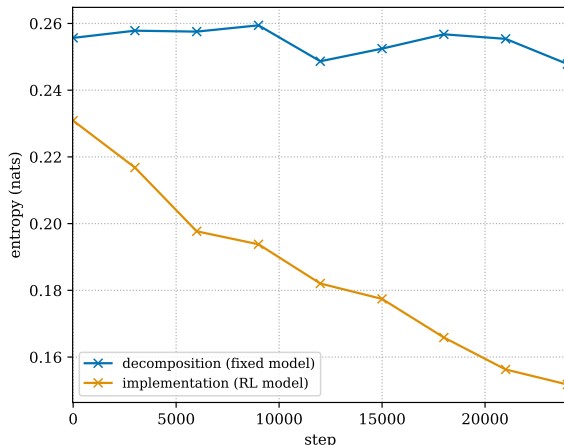

Figure 11: **Policy entropy of decomposition- and implementation-only training over the course of training. Left:** decomposition-only training, the implementation policy of a fixed model only changes marginally with the changing prompts from the decomposition model while decomposition policy entropy decreases. **Right:** implementation-only training; the implementation policy entropy decreases over the course of training.

- Soft Policy Optimization (SPO) (Cohen et al., 2025) - optimized for diverse policies.

- our own logmeanexp utility function with regular inference which optimizes the mean between pass@1 and pass@k.

We train all baselines until the reward plateaus and tune the hyperparameters based on the best pass@10 performance on LiveCodeBench (see Figure 12). We set 16 samples per prompt for all baselines, $k = 8$ for pass@$k$ and $\beta = 0.3$ in the logmeanexp. Since DecompRL has a higher sampling budget per problem its reward plateaus after 1 epoch whereas baselines plateau after 3 epochs on the same training set.

## 15 Hyperparameters and Additional Training Details

We train our models with the Adam optimizer (Kingma & Ba, 2015) ($\beta_1 = 0.9, \beta_2 = 0.95$) with decoupled weight decay (Loshchilov & Hutter, 2019) factor 0.1 and gradient clipping at a norm of 1. We warm up the learning rate linearly over 200 steps to a final value of $6 \times 10^{-8}$. We use a local batch size of 2. Trainers discard the first 10 batches at the beginning of a run as warmup.

Our distributed runs use 80 H100 GPUs, split up into 8 trainer GPUs and 72 worker GPUs. On the worker side, we use temperature 1.0 for generations. During decomposition training, we set the number of implementations $k = 8$, a maximum decomposition size of $n_{\max} = 6$ and conduct up to $m = 1024$ evaluations per decomposition. We run three separate decomposition trainings with $\beta = 0.03, 0.07, 0.1$ with Qwen 2.5 7B which according to Figure 3 corresponds to a mixture of pass@k objectives for $1 \leq k \leq 1000$. We pick $\beta = 0.07$ as it had the highest pass@10 performance on the DeepMind Code Contest validation set.

At evaluation time we use temperature of 1.0 and up to 1000 decompositions per problem with each up to 4096 evaluations.

**Code execution.** For all our code evaluations, we use a sandbox similar to the one provided for Li et al. (2022) using Python 3.11. We set a time limit of $2T + 1$ seconds for all test cases if $T$ is the number of test cases. We execute code on an external CPU cluster that parallelizes unit test executions. Each problem in our training set has a mean of 10 unit tests, each with average size $< 10^2$ bytes which we estimated to take $< 0.1$ seconds to execute in our evaluation setup. If we have 512 evaluations per problem during training, this gives an upper bound of $512 \times 0.1 = 51.2$ seconds per problem.

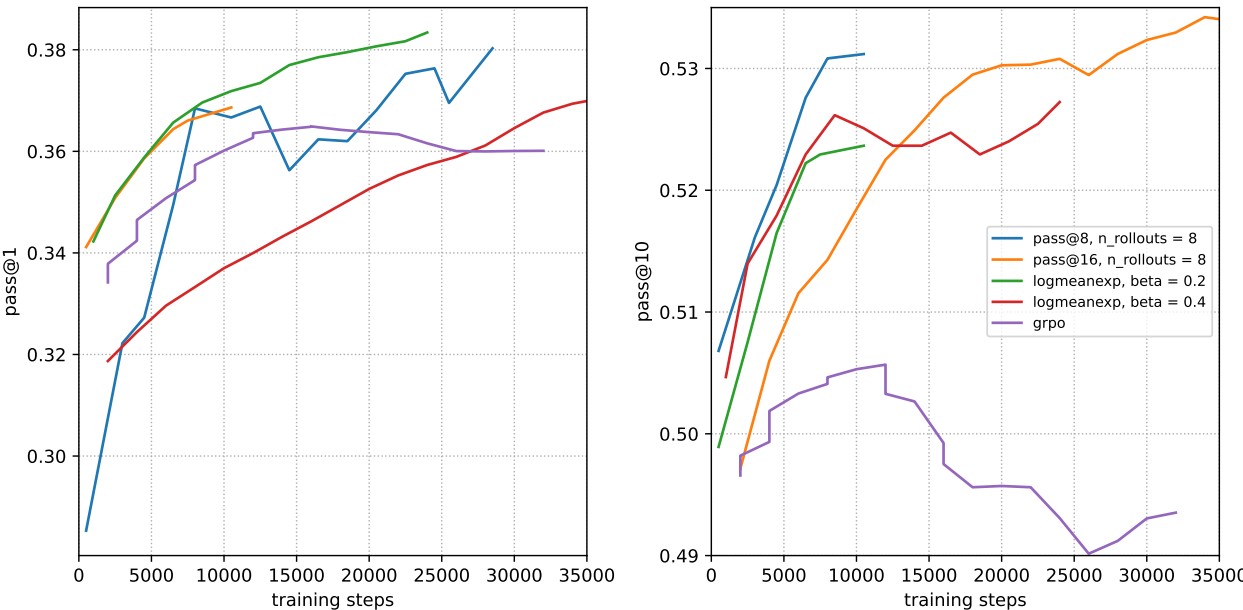

Figure 12: **Evaluation results during training for baseline methods** on the LiveCodeBench v5 split. Experiments ran with the Code World Model 32B FAIR CodeGen team et al. (2025). Sampling is performed with nucleus sampling with a probability mass of 0.95 and temperature 0.6 for all methods.

## 16 Prompts

We use language models as decomposition and implementation policies.

For decompositions, we use the following decomposition prompt with a one-shot example, where the {context} is the dataset's problem description:

---

**Decomposition Prompt**

```
To solve complicated reasoning problems, it is often helpful to break them down into components.
Can you help a student of computer science solve a competitive programming problem by
breaking it down into sensible component functions?

## Example problem
An inversion in an array is a pair of elements where the first element is greater than the second
element,
and the first element appears before the second element in the array.
For example, in the array [2, 4, 1, 3, 5], the inversions are (2, 1), (4, 1), and (4, 3). The
total number of inversions is 3.

Task: Write a function that counts the number of inversions in an array efficiently.
The goal is to achieve this in O(n log n) time complexity.

## Example decomposition into component functions
```python
def merge_and_count(left: list[int], right: list[int]) -> tuple[list[int], int]:
    """
    Merges two sorted lists and counts the number of inversions.
    Args:
        left (list[int]): The left sorted subarray.
        right (list[int]): The right sorted subarray.
    Returns:
        tuple[list[int], int]: A tuple containing the merged sorted list and
        the count of inversions found during the merge.
    """
    pass

def merge_sort_and_count(arr: list[int]) -> tuple[list[int], int]:
    """
    Recursively sorts an array and counts the number of inversions.
```

---

```
    Args:
        arr (list[int]): The array to be sorted and analyzed for inversions.
    Returns:
        tuple[list[int], int]: A tuple containing the sorted array and the
        total count of inversions in the array.
    """
    pass

def count_inversions(arr: list[int]) -> int:
    """
    Counts the number of inversions in an array using a modified merge sort.
    Args:
        arr (list[int]): The array to be analyzed for inversions.
    Returns:
        int: The total number of inversions in the array.
    """
    pass
```

In this example decomposition of the programming problem, we give the student the idea
to use count the number of inversions by modifying the merge-sort step of the merge-sort algorithm.
We explain clearly and concisely what the component functions are supposed to do by means of
their type signatures and their docstring, but leave the implementation to the student.

Now, let's do the same for the following competitive programming question:

## Student's competitive programming problem to decompose:
{context}

## Instructions
Please decompose the programming problem into component functions like in the example above.
Guidelines:
- Explain all the functions that the student will need to solve the problem without implementing
them.
- Annotate each function with a type signature for its inputs and outputs.
- For each function, write a clear and concise docstring describing what the function does.
- Do not implement the functions: put 'pass' as their body.
- The functions can call each other but they should "do one thing" only whereever possible.
  Their behavior should be fully described by their docstring.
- If shared state needs to be passed between the functions, describe it in one of their docstrings
   and make it an argument for the functions that need to read or modify it.
- Enclose the functions in a single code block using triple backticks like so: ```python YOUR CODE
 HERE ```.
- You can reason about the problem first before starting the code block.
```

After the decomposition step, we extract the function headers and documentation strings from the language
model's generation. We implement each function in the decomposition using the prompt below. We
have {decomposition} extracted from our first language model call, {current_name} the function being
implemented, and {other_names} other function descriptions in the decomposition.

**Implementation Prompt**

```
I'm trying to solve the following code competitive programming problem:
## Problem description:
{context}

## Instructions
I have decomposed the problem into the following component functions:
```python
{decomposition}
```

Please help me implement the function {current_name}.
You can assume the following functions have been implemented correctly and use them without
defining them in your code:
{other_names}

Guidelines:
- Enclose the code in triple backticks like so: ```python YOUR CODE HERE```.
- You can reason about the problem first before starting the code block.
- You can add import statements on top if necessary.

Now, please implement this
```python
```

```
{current_code}
```
by filling in the function body, keeping the signature and docstring.

## 17 Reward Tensor Structure

**TMTOWTDI?** [3] We investigate the structure of reward tensors. For this, we generate solutions to CodeContests validation problems using hierarchical inference with $d = 2$ decompositions of a maximal size of 6 and $k = 4$ implementations per function. We evaluate all resulting combinations, i.e. up to $4^6 = 4096$ per problem, and analyze the resulting reward tensors $r$ of mode (number of axes) $n$ and shape $k \times \ldots \times k$. Assume that each implementation is either correct or incorrect, at an individual level, and that combinations pass the tests if and only if all their constituting implementations are correct. In this case, correctness of implementations can be expressed as the maxima $m_i$ across all modes of $r$ except the $i$-th, and the reward tensor would equal the outer product $r^{(1)} = m_1 \otimes \ldots \otimes m_n$ of the $m_i$, which is a rank-1 approximation of $r$. More generally, the rank of a Boolean tensor using the Boolean operations ($\lor$ for addition, and $\land$ for multiplication) is known as the Boolean rank (de Caen et al., 1981). In the case of $n = 2$ it is also known as the bipartite dimension of the corresponding bipartite graph. It is NP-hard to compute(Garey & Johnson, 2002).

From 20 decompositions with at least one positive reward, we obtain the following statistics: In 17 cases, $r = r^{(1)}$ agrees with the rank-1 approximation. Overall, viewing $r^{(1)}$ as a prediction of $r$, there are 859 true positives, 483 true negatives, 34 false positives and no false negatives (the latter by definition).

## 18 Number of Tokens Generated

Figure 2, 4a, and Table 1 compare RL-trained models (SPO, GRPO, pass@8, logmeanexp) against our hierarchical inference method (DecompRL) for different token counts. We consider one attempt as one code generation per problem for standard inference and one full decomposition with 8 implementations, 4096 max evaluations for hierarchical inference. To ensure a fair comparison, we compare the pass rates per generated tokens which correspond to a different number of attempts for each RL algorithm. Table 2 summarizes the average cost of an attempt per method for the LLama 3.1 8B and Qwen 2.5 7B models.

| Model | RL Algorithm | Avg Tokens per Attempt |
|---|---|---:|
| Llama 3.1 8B Instruct | no training | 317 |
| | GRPO | 215 |
| | pass@8 | 942 |
| | SPO | 295 |
| | DecompRL | 10495 |
| Qwen 2.5 7B | no training | 387 |
| | GRPO | 545 |
| | pass@8 | 570 |
| | logmeanexp, 16 samples | 500 |
| | logmeanexp, 48 samples | 538 |
| | DecompRL | 4000 |

Table 2: **Token count per code attempt.** Overall RL training increases the the number of generated tokens per attempt. A DecompRL attempt encompasses 1 decomposition and $n \times 8$ implementations for $n$ functions so it contains much more tokens.

We compare success rates for a fixed budget of tokens similar to the pass@t from Ouyang et al. (2022)

---

[3]There's more than one way to do it?

## 19   Example DecompRL

We show below an example of a code problem extracted from LiveCodeBench and a generated decomposition with 6 functions from a Qwen 2.5 7B model trained with DecompRL. This decomposition leads to correct code solutions.

---

**Problem Description**

```
You are given a positive number n.

Return the smallest number x greater than or equal to n, such that the binary representation of x
contains only set bits

Example 1:
Input: n = 5 Output: 7
Explanation: The binary representation of 7 is \"111\".

Example 2:
Input: n = 10 Output: 15

Explanation: The binary representation of 15 is \"1111\".

Example 3:
Input: n = 3 Output: 3

Explanation: The binary representation of 3 is \"11\".

Constraints: 1 <= n <= 1000
```

---

**Generated Decomposition**

```
Certainly! Here's the decomposition of the problem into component functions, along with the type
signatures and docstrings for each function:

def next_power_of_two(n: int) -> int:
    pass

def set_bits_until_power_of_two(n: int) -> int:
    pass

def smallest_number(n: int) -> int:
    pass

def count_set_bits(x: int) -> int:
    pass

def main(n: int) -> int:
    pass

def smallestNumber(n: int) -> int:
    power_of_two = next_power_of_two(n)
    set_bits = set_bits_until_power_of_two(power_of_two)
    x = (power_of_two - 1) or set_bits
    return x

In this decomposition:
next_power_of_two finds the smallest power of two greater than or equal to n

set_bits_until_power_of_two counts the number of set bits in n until it reaches the next power of
two.

smallest_number uses these two functions to find the smallest number x with all set bits in its
binary representation.

count_set_bits simply counts the number of set bits in a given number x.

main is a placeholder for any additional logic.

smallestNumber is the main function that users will call, which leverages the other functions to
solve the problem.
```

## 20 Licenses of Used Datatsets

We use datasets with the following licenses:

- CodeContests (Li et al., 2022): CC BY 4.0 with MIT and Apache 2.0 licenses components.

- TACO (Li et al., 2023): Apache 2.0.

- LiveCodeBench (Jain et al., 2024a): CC BY 4.0 with MIT and Apache 2.0 licenses components.

