# OpenReview forum: "DecompRL: Solving More Problems with Less Tokens"
_TMLR — Decision pending for TMLR_

### Review · Reviewer_eag2 · 2026-04-04

**Summary Of Contributions:**

This paper proposes DecompRL, a reinforcement-learning framework for hierarchical code generation. Instead of generating a full program in an autoregressive manner, DecompRL first produces a decomposition of the programming problem into functions specified by signatures and docstrings, then generates multiple candidate implementations for each function independently, and finally recombines those implementations into many full candidate programs for execution and verification, to shift search bottleneck from expensive GPU-side generation to cheaper CPU-side evaluation in automatically verifiable domains such as competitive programming. The paper introduces an RL training recipe with separate decomposition and implementation policies, along with a recombination-based gradient estimator and a logmeanexp multi-sample objective intended to better optimize large-pass@k behavior. The paper also provides empirical evidence that this approach can outperform several standard RL baselines in the high-inference-budget regime on competitive-programming benchmarks. Across multiple benchmarks, including LiveCodeBench, CodeContests, and DeepMind Code Contest, DecompRL outperformed standard RL and diversity-oriented baselines, including GRPO, pass@8 training, SPO, logmeanexp training, especially in the high-token regime.

**Audience:**

Yes

**Audience Explanation:**

I expect this paper would interest at least several parts of the TMLR audience, such as researchers in RL for LLM, since the paper advances a concrete alternative to repeated whole-solution sampling by reusing partial generations combinatorially.

**Broader Impact Concerns:**

There are no broader impact concerns.

**Claims And Evidence:**

No

**Claims Explanation:**

The central empirical claim that the DecompRL can outperform standard RL baselines in the large-budget setting by exploiting decomposition and recombination is reasonably supported. The most complete result is Table 1 on overall LiveCodeBench with Qwen 2.5 7B, where DecompRL is not best at small budgets but becomes competitive or best at larger token budgets, reaching the strongest reported numbers at 50k, 100k, and 500k tokens.

However, I am concerned that some of the claims are stronger than actually supported by the evidence.

First, the evidence is entirely within code generation for the competitive programming domain, despite the abstract and introduction reading as though the method broadly addresses "complex problems" or "automatically verifiable problems." I think a large portion of the proposed method relies on the modular structure of code generation tasks, especially in the competitive programming domain, where the generated code is not very long, so unless there is enough evidence that DecompRL's idea can be useful for other domains or tasks, these are overclaiming the applicability. I would recommend reframing the abstract and introduction to accurately describe the applicability.

Second, the presentation around Figure 1 says DecompRL can solve harder problems than other methods, but it is not clear how it is defined. Specifically, I could not find a clear explanation of how the easy/medium/hard subsets are defined for LiveCodeBench, making the difficulty-based framing harder to evaluate. Moreover, Table 1 shows gains mainly at large token budgets, and the limitations section explicitly states that DecompRL performs worse than the starting policy for Qwen 2.5 7B on the easy split of LiveCodeBench, so the headline claim should be more accurate.

Lastly, the key systems claim that the DecompRL shifts the bottleneck from GPU inference to CPU evaluation, which needs to be quantified. While the reasoning that generating $K^N$ unique candidate solutions from the token budget of $K \times N$ shifts the bottleneck from GPU to CPU, it would be valuable to quantify the claim, such as via wall-clock latency evaluation of GPU inference vs. CPU verification.

**Requested Changes:**

Major points:
1. Please narrow and clarify the framing in the abstract and introduction.
2. Please define how LiveCodeBench is split into easy/medium/hard subsets.
3. Please provide quantified evidence that DecompRL shifts the RL bottleneck from GPU-based inference to CPU-based.

Minor points:
1. It would be interesting to see more analysis of decomposition quality, on top of the decomposition size. The paper shows that larger decompositions can correlate with better high-budget performance, but size is only a very high-level proxy. For example, it would strengthen the paper to analyze whether good decompositions have any characteristics in terms of interfaces, redundancy, or semantic modularity.
2. It would be interesting to show more data about the robustness of DecompRL to out-of-distribution problems in terms of decomposition structure. For example, how well does the fine-tuned model perform for problems that require different program lengths or numbers of functions than those seen during training?

---

> ### Author Response · Authors · 2026-05-12
> **Answer to Reviewer eag2**
>
> We thank the reviewer for the careful read and for the framing of our work as "a concrete alternative to repeated whole-solution sampling by reusing partial generations combinatorially". We appreciate the detailed suggestions on how to improve this draft and address them below:
>
> Major points:
> 1. See shared answer to both reviewers.
> 2. LiveCodeBench easy/medium/hard splits. Thank you for catching this, we will add an explicit description in Section 3 and in the Figure 1 caption. The splits we use are the official problem-difficulty annotations released with LiveCodeBench (Jain et al., 2024a), which are propagated from the source competitive-programming platforms (LeetCode, Codeforces, AtCoder). For the 2024/08/01–2025/02/01 range, this gives 109 easy, 110 medium and 60 hard problems.
> 3. See shared answer to both reviewers.
>
> Minor point:
> 1. Decomposition analysis. Thank you for the suggestion, we did an analysis of 8000 decomposition sampled from Qwen 2.5 7B training runs to find correlations between high reward and other characteristics. We found beyond size, shorter and simpler decompostions do better:
> - Functions with few parameters (1-2 max) rather than general vague functions
> - Full typed interface (input/output type specified in the docstring) achieved +18% success rates than ones were the types were missing.
> - Less verbose docstrings: on average shorter function docstrings correlated with higher success (r = 0.3)
> We will add this to section 4.2.
>
> 2. OOD tasks: Our reported LiveCodeBench numbers are already out of distribution with respect to the training distribution: we train on CodeContests/TACO and evaluate on LiveCodeBench (Section 3.1), which differs in problem style, length, and decomposition structure. We chose LiveCodeBench specifically because it was the best non-contaminated benchmark we could find for our base models (Qwen 2.5 7B, Llama 3.1 8B, CWM 32B), with a release window post-dating their pretraining cutoffs. That said, we are very open to suggestions if the reviewer has other code benchmarks in mind that would better stress OOD decomposition structure and that are clean with respect to our models' pretraining.

---

### Review · Reviewer_xarB · 2026-04-07

**Summary Of Contributions:**

This paper proposes DecompRL, a reinforcement learning framework for code generation that trains a model to first decompose a programming problem into a set of functions and then generate multiple implementations for each function independently. By recombining these implementations, the method can produce many complete candidate programs while using relatively few language-model forward passes, thereby shifting the scaling bottleneck from GPU generation to CPU-based evaluation. The method uses two policies (decomposition and implementation), a recombination-based policy gradient estimator, and a logmeanexp-based multi-sample objective designed for large-pass@k regimes. Experiments on CodeContests and LiveCodeBench show gains over several RL baselines when the evaluation budget is very large.

**Additional Comments:**

The main strengths are:
1. The paper tackles an interesting and underexplored question: how to train models for combinatorial recombination rather than only improving single-sample quality.
2. The method is conceptually nontrivial: the decomposition/implementation factorization, recombination-based credit assignment, and logmeanexp objective form a coherent framework.
3. The empirical section includes not only benchmark numbers but also mechanism-oriented analyses and ablations, including the effects of recombination, decomposition size, sequential training, and leave-one-out baselines.

The main weaknesses are:
1. The practical scope appears narrow: the method relies on domains where one can cheaply and reliably evaluate very large numbers of candidate solutions.
2. The evaluation focuses heavily on extremely large pass@k regimes, where the real-world value of the metric becomes questionable.
3. The training/inference setup is fairly specialized and resource-intensive, which weakens the practical appeal of the proposed compute tradeoff.

Overall, I find the paper technically interesting and reasonably well executed, but I have substantial concerns about the practical relevance of the setting and the reliance on very large-pass@k evaluation. These concerns currently prevent me from being more enthusiastic.

**Audience:**

Yes

**Audience Explanation:**

The paper would interest readers working on reinforcement learning for LLMs, inference-time scaling, code generation, and verifier-based training.

**Broader Impact Concerns:**

I do not have major ethical concerns beyond the standard ones associated with large-scale code generation. One possible issue is that methods optimized for large-budget search could make it easier to generate large numbers of candidate programs efficiently, which may amplify both benign and harmful use cases. However, I do not see this as a central blocker for publication.

**Claims And Evidence:**

Yes

**Claims Explanation:**

Within the benchmark setting studied in the paper, the empirical evidence is generally consistent with the main claims. The paper clearly explains the hierarchical generation setup, the recombination mechanism, the policy-gradient formulation, and the multi-sample objective. The experiments show that DecompRL improves performance relative to several baselines when the evaluation budget is pushed to very large scales, and the ablations support the importance of logmeanexp aggregation, leave-one-out credit assignment, and sequential training.

**Requested Changes:**

Critical concerns

1. Clarify the practical scope of the method much more explicitly. The paper should better delimit where DecompRL is expected to matter in practice. Its advantages depend on a regime with cheap, reliable, massively parallel evaluation of candidate solutions. This is true for competitive programming and some verifier-heavy domains, but much less true for many realistic coding or reasoning applications. The current framing occasionally reads as if the method were broadly applicable to “harder problems” in general, whereas the evidence mainly supports a narrow class of automatically verifiable tasks.
Address the practical meaning of very large pass@k.

2. A central concern for me is that the paper optimizes and evaluates mainly in very high-pass@k regimes. I would like the authors to discuss much more carefully why performance at very large
k should be considered meaningful in practice, rather than mainly a search-quality proxy. As it stands, the work can be read as improving “eventual search success under huge budgets” rather than improving solution quality under realistic latency or compute constraints.

3. Include stronger evaluation at realistic operating points.
Since the paper itself acknowledges that DecompRL is not expected to improve pass@1, and may even degrade easier or low-budget settings, I think the empirical section would be stronger if it emphasized realistic small-k or limited-budget tradeoffs much more directly. At minimum, the paper should more clearly characterize the crossover point where the method starts to become beneficial.

4. Provide better end-to-end cost accounting.
The method is motivated as shifting compute from GPU inference to CPU evaluation, but the overall system still uses substantial infrastructure, including large-scale asynchronous RL training and external CPU resources. I would like a clearer accounting of wall-clock cost, hardware requirements, and whether the gains remain attractive when these are included.

Changes that would strengthen the work

5. Better analyze credit-assignment quality under partial recombination sampling.
Since training evaluates only a subset of the possible recombinations, the action-level credit assignment is necessarily approximate and may have nontrivial variance depending on coverage. A more explicit discussion or analysis of this issue would improve the paper.

---

> ### Author Response · Authors · 2026-05-12
> **Answering the questions of Reviewer xarB**
>
> Thank you for finding our paper interesting and well executed, we appreciate your detailed questions and suggestions regarding our work. We address your critical concerns below:
> 1. See shared answer to both reviewers.
> 2. **Focusing on high pass@k regime:** We entirely agree that performance at low pass@k matters and an interesting follow up to our work would be distilling the DecompRL performance into a policy’s pass@1 behavior (via Supervised Fine Tuning for eg). We also argue focusing on high pass@k is extremely meaningful for three main reasons:
> - Inference scaling allows models to solve very hard problems not solvable at low budgets. This is the case for 7B models on benchmarks like LCB, but it is also the primary mechanism state-of-the-art models use to push the frontier in complex mathematics (Trinh et al., 2024).
> - Frontier reasoning-model training pipelines (Guo et al., 2025; Rastogi et al., 2025; Team Kimi, 2025) increasingly rely on a two-stage loop: a strong "explorer" model performs large-scale inference-time search to discover verified-correct trajectories on hard problems, which are then distilled via SFT or pass@1 RL into a fast, low-latency student. The explorer stage is fundamentally bottlenecked by success rate at very large k: a model cannot teach what it has never solved. DecompRL targets exactly this stage: by raising pass@k on the hardest problems, it widens the set of tasks for which any correct trajectory can be harvested, directly enlarging the training distribution of the eventual pass@1 model.
> - High-Assurance Applications and Automated Oracles: In domains like formal verification, automated program repair, and cryptography, the cost of failure is immense, but verifying a solution is often cheap and deterministic (e.g., using compilers or formal provers) (First et al., 2023; Liu et al., 2023). In these settings, exhaustively generating a massive number of candidates to find a single, provably correct solution is a practical and highly valuable deployment strategy.
> 3. Thanks for the suggestion, we can add examples of DecompRL vs. standard generations at realistic small-k regimes. The crossover point depends on the model strength (Figure 4a) and problem difficulty (Figure 1), we can add a graph showing how it decreases with model strength on a fixed dataset.
> 5. The leave-one-out baseline used in Equation 2 (analysed in Appendix 12) is precisely designed to control variance under partial recombination: we show that its Signal-to-Noise-Ratio (higher is better) scales as O(n) versus O(1/n) for the naive estimator. Although we only sample a sparse subset of the 8^6 combinations, the reward-tensor analysis in Appendix 17 shows that for most decompositions (17/20 in the analyzed set), the joint reward matrix is well-approximated by the rank-1 outer product of per-implementation correctness indicators. This factorized structure means each implementation's contribution can be reliably estimated from a small fraction of the recombinations, because the credit signal is concentrated along the function axes rather than spread across joint interactions. We will add a paragraph in Section 2.4 making this point explicit.
>
> Regarding the concern that _"the training/inference setup is fairly specialized and resource-intensive, which weakens the practical appeal of the proposed compute tradeoff."_, we agree that the setup is ressource-intensive however the async RL framework we use is now standard for LLM post-training [Copet et al., 2025; Rastogi et al., 2025; Gehring et al., 2024]. DecompRL only adds a different rollout strategy and expert-iteration training on top. We also argue that DecompRL's inference is simpler than other competing search methods (tree search, MCTS, self-refinement) and that a cheaper more CPU-heavy cluster would actually amplify its gains (see CPU/GPU trade-off in the shared answer.)
>
> We will make sure to emphasise these points in the final draft and reduce the framing of the paper.
>
> Sources:
> - Trinh, Trieu H., et al. "Solving olympiad geometry without human demonstrations." Nature 625.7995 (2024): 476-482.
> - Guo, Daya, et al. "Deepseek-r1: Incentivizing reasoning capability in llms via reinforcement learning." arXiv preprint arXiv:2501.12948 (2025).
> - Rastogi, Abhinav, et al. "Magistral." arXiv preprint arXiv:2506.10910 (2025).
> - Team, Kimi, et al. "Kimi k2: Open agentic intelligence." arXiv preprint arXiv:2507.20534 (2025).
> - First, Emily, et al. "Baldur: Whole-proof generation and repair with large language models." Proceedings of the 31st ACM Joint European Software Engineering Conference and Symposium on the Foundations of Software Engineering. 2023.
> - Copet, Jade, et al. "Cwm: An open-weights llm for research on code generation with world models." arXiv preprint arXiv:2510.02387 (2025).
> - Gehring, Jonas, et al. "Rlef: Grounding code llms in execution feedback with reinforcement learning." arXiv preprint arXiv:2410.02089 (2024).

---

### Review · Reviewer_38Y9 · 2026-05-17

**Summary Of Contributions:**

**Contributions:**
1. DecompRL: hierarchical RL training two policies (decomposition π(D), implementation π(Iᵢ|D)) for modular code gen
3. Recombination: nk forwards → kⁿ evaluable trajectories (GPU→CPU bottleneck shift)
3. logmeanexp_β advantage: smooth interp. between mean & max; approximates avg of pass@1/10/100/1000
4. Variance reduction thm (Thm 9.2) via Hoeffding decomp.
5. Sequential training w/ LOO baseline for credit assignment
6. Gains at high token budgets (≥50k/problem) on LCB-hard & CodeContests across 3 models

**Strengths:** novel objective; theoretical grounding; clear regime of applicability; thorough ablations; honest limits section

**Weaknesses:** underperforms baselines at low/mid budgets (contradicts "less tokens" title); cond. indep. assumption weakly validated; missing Parsel/MapCoder inference-only baselines at matched compute; 1 seed, no CIs; typos throughout

**Additional Comments:**

logmeanexp ↔ pass@k correspondence (Fig 3, Appx 11) is nice. maybe promote to main text?

Appx 9 proof clean

Fig 4 subplots small, labels hard to read; align Fig 5b,c axes

**Audience:**

Yes

**Audience Explanation:**

1. Sits at intersection of inference-scaling, RL-for-code, modular generation ...  very active areas
2. Recombination as RL signal (not just inference trick) is genuinely new vs Parsel/CodeChain
3. logmeanexp + Hoeffding-variance analysis reusable beyond code
4. Negative result (hurts at low budgets / easy problems) informative
5. Audience: RL-for-LLM, code-gen, expert-iteration researchers

**Broader Impact Concerns:**

Worth a short paragraph on: (i) decomp+recomb extension to dual-use code (e.g., exploit synthesis), (ii) compute asymmetry / method advantages actors w/ large CPU clusters

**Claims And Evidence:**

No

**Claims Explanation:**

- Title/abstract "Less Tokens" contradicted by Table 2 (10495 vs 215–942 tok/attempt) & Table 1 (DecompRL loses to GRPO/lme16 below 50k budget)
- No seed variance / CIs; many reported deltas <0.04 on ~100-problem benchmarks
- RL contribution not isolated from hierarchical-inference contribution (no Parsel-style inference-only baseline on same SFT ckpt)
- Cond. indep. assumption justified only by §17's 20-decomposition study on one dataset
- Fig 1 LCB-hard: dashed/extrapolated regions unclear

**Requested Changes:**

- Reframe title/abstract — "Less Tokens" is inaccurate; state the regime (≥50k tok, hard problems) where gains hold; acknowledge sub-50k underperformance
- Report seed variance / bootstrap CIs for Table 1 & Fig 4a
- Add Parsel-style inference-only baseline on same SFT ckpt at matched budget to isolate RL vs hierarchical-inference contributions
- Strengthen cond. indep. validation (§17): more datasets, more decomps, failure-rate vs n
- Clarify Fig 1: measured vs extrapolated, n_seeds, error bars
- Fix typos & notation: "off-polyciness", "Surpressing", "worst"→"worse", inconsistent k/h in §2.3 standard estimator, suspicious or in §19 generated code

Optional (but appreciated):
- Match-compute comparison vs MapCoder/CodeChain/D&C
- Report wall-clock & CPU-hours to quantify "CPU is cheap"
- Resolve Fig 4c (size↓) vs Fig 5c (larger size→better) tension
- Empirically demonstrate the claimed DecompRL→SFT distillation loop
- Disambiguate Fig 6 entropy axes (decomp/impl/joint?)
- Rewrite §2.4 decomp policy gradient indexing ({r_{i'j} | i' ≠ i} unclear)
- Qualitative section: when does decomposition help vs hurt?
- Memory cost of two-model setup
- §10 (Gumbel view) feels orphaned — tie to method or move
- Release trained decomposition checkpoints
- Remove footnote 3 ("TMTOWTDI?") please ... it's unprofessional

---

> ### Author Response · Authors · 2026-05-28
> **Answer to Reviewer 38Y9**
>
> We thank the reviewer for the thorough review. We address each requested change below.
> 1) We agree "Less Tokens" is misleading. DecompRL's advantage is at high budgets where recombination yields ~50× GPU-token reduction (see shared response). We propose the title **"DecompRL: Solving Harder Problems by Learning Modular Code Generation"** and will reframe the abstract accordingly.
> 2) We agree this would strengthen the paper but did not include seed variance because each RL training run is compute-intensive, requiring 2–3 days on 80 H100 GPUs for the smallest model (Qwen 2.5 7B, see Appendix 15). Given this constraint, we prioritized validating our method across different model scales (Llama 3.1 8B, Qwen 2.5 7B, CWM 32B) and benchmarks (LiveCodeBench, CodeContests) rather than running multiple seeds at a single scale. However, bootstrapping CIs on the evaluations is feasible: each evaluation run takes approximately 20 minutes with TP=4 on 32 GPUs for Qwen 2.5 7B (we will add these wall clock time details in appendix). Thanks for the suggestion, we will add this to the final version by evaluating the last 3 training checkpoints with multiple sampling seeds. We do note that the high-budget regime where DecompRL shines is precisely where the reported numbers are most reliable (since variance decreases with sampling size).
> 3) Our hierarchical inference is simpler than Parsel (Zelikman et al., 2023) — no intermediate language or generated tests. Models pay a format tax initially, but RL recovers it: decomposition RL adds +8% pass rate, implementation RL +2%, and parsing success rises 30%→95%.
> | Method | pass@1 (~4k tok) |
> |---|---|
> | Qwen 2.5 7B — Standard | 14.0% |
> | Qwen 2.5 7B — Hierarchical (no RL) | 11.0% |
> | Qwen 2.5 7B — DecompRL | 18.0% |
> | CWM 32B — Hierarchical (no RL) | 11.9% |
> | CWM 32B — DecompRL | 27.8% |
> We will add these baselines to Table 1.
> 4) Thank you for finding Appendix 17 interesting, we originally explored this to come up with a better sampling strategy than uniform sampling but we do agree it can add overall to the paper and will extend this to 100 decompositions (~4 days go run) and report rank 1 % vs. size of decomposition.
> 5) All Figure 1 curves are measured, not extrapolated.
> 6) Thank you for catching these we appreciate the detailed read and will fix these.
>
> **Optional changes (our responses):**
> **Match-compute comparison vs MapCoder/CodeChain/D&C.**:  We agree this comparison would be informative. We note that we deliberately focused our baselines on methods optimized for high-diversity RL (GRPO, SPO, pass@k, lme), since the core novelty of DecompRL lies in the RL training recipe rather than the decomposition idea itself; which was is a simplified version of prior work (Parsel, Zelikman et al., 2023).
>
> **Wall-clock & CPU-hours.** Addressed in the shared response.
>
> **Resolve Fig 4c vs Fig 5c.** Thank you for pointing this out, we will make sure to clarify this in section 4. Fig 5c shows that at evaluation time, larger decompositions scale better with budget because more functions means more recombinations (k^n grows with n). Fig 4c shows that during training, the policy nevertheless learns to shrink decomposition size. This happens because smaller decompositions have higher per-sample success rates (fewer functions that all need to be correct), so the RL objective rewards them in the short term even though they sacrifice the long-term scaling advantage. We identify this as a form of reward hacking (Section 5, "Size collapse"): the model takes the easy path of producing fewer, simpler functions rather than the harder path of improving the quality of larger decompositions. We use lower β values in logmeanexp to delay this collapse (Fig 4c).
> **DecompRL→SFT distillation loop.** We agree this would be a compelling demonstration and will add a concrete experimental protocol in the future work section.
>
> **Disambiguate Fig 6 entropy axes.** The ablations in Figure 6 were all run during the decomposition policy training since it is the first policy trained in our sequential loop and, as our hierarchical inference results confirm (see point 3 above), the most important contributor to DecompRL's performance. We will specify this in the caption.
>
> **Qualitative section** Problem types with natural sequential structure (parse → transform → compute → output) where sub-functions can be implemented and verified independently tend to benefit most from DecompRL. We will add examples in the appendix.
>
> **Memory cost.** Model weights account for less than 10% of available GPU memory on the workers (~7 GiB vs ~3.5 GiB per GPU for two models vs one for the Qwen 2.5 7B). Importantly, only the workers bear this extra cost as the trainer is unaffected since it only trains one model. In practice, we observed zero OOMs and no need for CPU offloading throughout training even with CWM 32B. The overhead is comparable to that of hosting an LLM-as-a-judge on workers, a common practice in RLHF pipelines.

---

### Author Response · Authors · 2026-05-12
**Addressing major concerns of all reviewers**

We thank the reviewers for their detailed and constructive feedback. We appreciate the enthusiasm regarding our method, as an alternative to whole-solution RL. Both reviewers raise overlapping concerns about (i) the practical scope and framing, (ii) quantification of the GPU→CPU bottleneck shift. We address these in a shared response below:

**Narrowing and clarifying the framing.** We agree and will revise the abstract and introduction to explicitly scope DecompRL to automatically and cheaply verifiable tasks where solutions can be hierarchically decomposed into independently solvable sub-parts. This naturally fits competitive programming, but as we now explicitly note, many AI for science tasks as well such as: (a) automated theorem proving where each lemma/sub-goal is independently checkable in Lean (Lample et al., 2022; Gloeckle et al., 2024); (b) Hardware Description Language for chip design where module graphs decompose into leaf modules each verifiable via unit tests (eg: RTLLM benchmark, Openllm-rtl, VerilogEval); (c) Olympiad-level mathematics with code-based verification (e.g., Project Euler-style problems by Haller et al., 2024); (d) reproducing scientific papers (e.g., Curie style benchmarks by Cui et al., 2025). We will rewrite the abstract and Section 1 to remove generic language about "complex problems" and to clearly describe the structural assumption (modular, verifiable trajectories). We will also retitle the conclusion subsection and limits section accordingly.

**Quantifying the GPU→CPU bottleneck shift.** Thank you for pointing this out, we will make sure to include a new section in the final draft that makes this explicit. Combining the per-attempt token costs in Table 2 (Appendix 18) with the cluster setup in Appendix 15 (72 worker GPUs, 8 trainer GPUs, external CPU cluster, ~10 unit tests per problem at <0.1s each), one DecompRL attempt costs ~4k generated tokens and yields up to 512 recombined candidates evaluated on CPU in at most ~51.2s per problem. Reaching the same 512 candidates with standard whole-code generation would instead require ~512 × 387 ≈ 200k tokens of GPU generation per problem. DecompRL therefore provides a ~50x reduction in GPU tokens for the same number of evaluated solutions.

In practice, let’s break down the wall clock cost per problem in our experimental setup with the Qwen 2.5 7B model:

*Standard (16 samples)*
- 10k generation tokens, 108s serial generation time, 16 remote execution calls,
- 371s serial execution time, 1.1s training cost, 479s total worker-time,
- 23\% GPU utilization, and 77\% CPU utilization.

*Standard (512 samples)*
- 198k generation tokens, 2100s serial generation time, 512 execution calls
- 11878s serial execution time, 21.8s training cost, $\sim$14000s total worker-time,
- 15\% GPU utilization, and 85\% CPU utilization.

*DecompRL (512 evals):*
- 4k generation tokens, 42s serial generation time, 512 execution calls,
- 11878s serial execution time, 0.4s training cost, $\sim$11900s total worker-time,
- 0.4\% GPU utilization, and 99.6\% CPU utilization.

When parallelized, the wall-clock saving from DecompRL scales with how GPU-bound the system is: at 8 GPUs with 128 remote execution threads, where generation dominates, DecompRL delivers a 3.6× speedup; at 8 GPUs with 64 threads, a 2.3× speedup; while at 72 GPUs with 128 threads, where remote execution already dominates, the saving shrinks to 1.3×. Since our cluster is heavily GPU-optimized, we argue scaling the number of CPUs (which is often cheaper than scaling GPUs) could unlock even larger gains for DecompRL since the marginal cost of evaluating more recombinations is essentially free on the GPU side.

Sources:
- Lample, Guillaume and Lacroix, Timothee and Lachaux, Marie-Anne and Rodriguez, Aurelien and Hayat, Amaury and Lavril, Thibaut and Ebner, Gabriel and Martinet, Xavier. “HyperTree Proof Search for Neural Theorem Proving” Advances in Neural Information Processing Systems, 2022
- Haller, Patrick, Jonas Golde, and Alan Akbik. "Pecc: Problem extraction and coding challenges." Proceedings of the 2024 Joint International Conference on Computational Linguistics, Language Resources and Evaluation (LREC-COLING 2024). 2024.
- Gloeckle, Fabian, et al. "ABEL: Sample efficient online reinforcement learning for neural theorem proving." The 4th Workshop on Mathematical Reasoning and AI at NeurIPS'24. 2024.
- Liu, Shang, et al. "Openllm-rtl: Open dataset and benchmark for llm-aided design rtl generation." Proceedings of the 43rd IEEE/ACM International Conference on Computer-Aided Design. 2024.
- Liu, Mingjie, et al. "Verilogeval: Evaluating large language models for verilog code generation." 2023 IEEE/ACM International Conference on Computer Aided Design (ICCAD). IEEE, 2023.
- Cui, Hao, et al. "Curie: Evaluating llms on multitask scientific long context understanding and reasoning." arXiv preprint arXiv:2503.13517 (2025).